# Conservative Offline Goal-Conditioned Implicit V-Learning

**Kaiqiang Ke** [1]  **Qian Lin** [1]  **Zongkai Liu** [1 2]  **Shenghong He** [1]  **Chao Yu** [1 3]

## Abstract

Offline goal-conditioned reinforcement learning (GCRL) learns a goal-conditioned value function to train policies for diverse goals with pre-collected datasets. Hindsight experience replay addresses the issue of sparse rewards by treating intermediate states as goals but fails to complete goal-stitching tasks where achieving goals requires stitching different trajectories. While cross-trajectory sampling is a potential solution that associates states and goals belonging to different trajectories, we demonstrate that this direct method degrades performance in goal-conditioned tasks due to the overestimation of values on unconnected pairs. To this end, we propose Conservative Goal-Conditioned Implicit Value Learning (CGCIVL), a novel algorithm that introduces a penalty term to penalize value estimation for unconnected state-goal pairs and leverages the quasi-metric framework to accurately estimate values for connected pairs. Evaluations on OGBench, a benchmark for offline GCRL, demonstrate that CGCIVL consistently surpasses state-of-the-art methods across diverse tasks.

## 1. Introduction

Goal-Conditioned Reinforcement Learning (GCRL) aims to train RL agents that are capable of mastering a diverse range of skills, each defined by a distinct goal (Schaul et al., 2015; Liu et al., 2022). In offline settings, these goals are typically identified as states already contained in the dataset, eliminating the need for manual goal design and facilitating the acquisition of general-purpose policies without active environment interactions. Such a setup is especially beneficial for real-world applications, where direct interactions can be costly, time-consuming, or even hazardous (Kumar et al.,

2020). Consequently, offline GCRL has attracted significant attention in recent years, enabling the learning of skillful agents from pre-collected datasets while circumventing the challenges of online data collection (Ma et al., 2022b; Park et al., 2023; Kim et al., 2024).

A common approach in offline GCRL is to train a goal-conditioned value function $V(s, g)$ from a static dataset via temporal difference (TD) learning (Sutton & Barto, 2018; Schaul et al., 2015; Ghosh et al., 2023). One of the primary challenges in value learning is the low learning efficiency issue caused by sparse rewards. Existing methods utilize Hindsight Experience Replay (HER) (Andrychowicz et al., 2017), which samples states and goals within a single trajectory, to tackle this problem. However, such approaches ignore the issue of the **cross-trajectory** state-goal pairs, i.e., pairs that can only be connected by stitching multiple trajectories. Such cross-trajectory pairs are essential in goal-stitching tasks, where the initial state and the goal could belong to different trajectories.

However, directly incorporating cross-trajectory state-goal pairs into data sampling can lead to value estimation errors, particularly when some of these pairs are **unconnected**—that is, when no valid sequence of transitions exists within the dataset linking the state to the goal. Therefore, their values cannot be estimated accurately using TD learning and tend to be consistently overestimated during training. Moreover, these overestimated values propagate to neighboring states through the bootstrapping process of TD learning, thus misleading the policy to select suboptimal actions.

In this paper, we investigate the above issue in depth. First, by theoretically analyzing the discrepancy between the estimated and optimal values, we show that the overestimation of values on unconnected state-goal pairs leads to propagation errors in adjacent states. Then, we propose **C**onservative **G**oal-**C**onditioned **I**mplicit **V**alue **L**earning (**CGCIVL**), a novel algorithm designed to learn a conservative estimate of the goal-conditioned value function. Specifically, a regularization term is incorporated to prevent overestimated values of unconnected pairs and a quasimetric model is employed to prevent potential under-estimation on cross-trajectory but connected state-goal pairs. Theoretical guarantees further show that the proposed method alleviates overestimation of values on unconnected pairs while

---
[1]School of Computer Science and Engineering, Sun Yat-sen University, Guangzhou, China [2]Shanghai Innovation Institute, Shanghai, China [3]Pengcheng Laboratory, Shenzhen, China. Correspondence to: Chao Yu <yuchao3@mail.sysu.edu.cn>.

*Proceedings of the $42^{nd}$ International Conference on Machine Learning*, Vancouver, Canada. PMLR 267, 2025. Copyright 2025 by the author(s).

maintaining accurate value estimates for all connected pairs. Empirically, experiments on OGBench (Park et al., 2024), a benchmark specifically designed for offline GCRL, demonstrate that our algorithm consistently matches or surpasses state-of-the-art methods across distinct environments with varying configurations.

## 2. Preliminaries

**Problem setting** The problem of offline GCRL can be formulated as a Goal-Augmented Markov Decision Process (GA-MDP) $\mathcal{M} = (\mathcal{S}, \mathcal{A}, T, R, \mathcal{G}, \gamma)$ and a dataset $\mathcal{D}$, where $\mathcal{S}$ denotes the state space, $\mathcal{A}$ denotes the action space, $T : \mathcal{S} \times \mathcal{A} \to \mathcal{S}$ denotes the transition function, $R : \mathcal{S} \times \mathcal{G} \to [r_{\min}, 0]$ denotes the reward function, $\mathcal{G}$ denotes the goal space, and $\gamma \in (0, 1]$ is the discount factor. The dataset $\mathcal{D}$ consists of trajectories $\tau = (s_0, a_0, s_1, a_1, \ldots, s_T)$. We consider any state in the dataset as a potential goal and define the goal space $\mathcal{G}$ as $\{s \mid s \in \mathcal{D}\}$.

The objective of offline GCRL is to learn a policy $\pi$ that maximizes the expected cumulative reward:

$$J(\pi) = \mathbb{E}_{\substack{g \sim \mathcal{G}, s_0 \sim \mu_0(\cdot), \\ a_t \sim \pi(\cdot|s_t,g)}} [\sum_{t=0}^{T-1} \gamma^t R(s_t, g)]. \quad (1)$$

**Off-policy value estimation in GCRL** In GCRL, the goal-conditioned value function is learned through an iterative process based on the optimal Bellman operator $\mathcal{T}$:

$$V_{k+1} = \min_V \mathbb{E}_{\substack{s \sim \mathcal{D}, \\ g \sim p_m^\alpha(g|s)}} [V(s, g) - \mathcal{T} V_k(s, g)], \quad (2)$$

where $\mathcal{T}$ is defined as:

$$(\mathcal{T} V)(s, g) := R(s, g) + \gamma \left[ \max_{a, s' \sim T(\cdot|s,a)} V(s', g) \right]. \quad (3)$$

The mixture distribution $p_m^\alpha(g \mid s)$ is defined as:

$$p_m^\alpha(g \mid s) := \alpha p_{\text{traj}}(g \mid s) + (1 - \alpha) p_{\text{rand}}^{\mathcal{D}}(g), \quad (4)$$

where $\alpha \in [0, 1]$ is the weight of the mixture and the components are defined as follows:

- $p_{\text{traj}}(g \mid s)$: Given a state $s_t$ at time $t$ in trajectory $\tau = (s_0, a_0, s_1, \ldots, s_N)$, a future state $g = s_m$ is sampled by selecting $m$ uniformly at random from the discrete interval $\{t, t+1, \ldots, N\}$.

- $p_{\text{rand}}^{\mathcal{D}}(g)$: A uniform probability distribution over all states in the dataset $\mathcal{D}$.

**Quasimetric model** Given a set $\mathcal{S}$, a quasimetric is a function $d : \mathcal{S} \times \mathcal{S} \to \mathbb{R}_{\geq 0}$ satisfying the following properties:

$$\forall s_1, s_2, s_3 \in \mathcal{S}, \quad d(s_1, s_2) + d(s_2, s_3) \geq d(s_1, s_3), \quad (5)$$
$$\forall s \in \mathcal{S}, \quad d(s, s) = 0. \quad (6)$$

Quasimetrics generalize metrics by relaxing the symmetry requirement. The space of all quasimetrics over $\mathcal{S}$ is denoted as $\mathcal{Q}(\mathcal{S})$, and its negation is defined as $\mathcal{Q}^-(\mathcal{S}) := \{-d | d \in \mathcal{Q}(\mathcal{S})\}$.

The quasimetric model $d_\theta$ is a parameterized model that satisfies the properties of the quasimetric, where $\theta$ is the parameter to be optimized. Specifically, a quasimetric model $d_\theta$ typically consists of (1) a deep encoder mapping two states $s_1, s_2 \in \mathcal{S}$ to $x_1, x_2$ in the latent space $\mathbb{R}^d$ and (2) a differentiable latent quasimetric head $d_{\text{latent}} \in \mathcal{Q}(\mathcal{S})$ that computes the quasimetric distance such as $||x_1 - x_2||_2$, for two input latents. In this work, we implement quasimetric models using Interval Quasimetric Embeddings (IQE) (Wang & Isola, 2022a), with details provided in Appendix C.1.

The optimal goal-conditioned value function under GA-MDP can be represented by quasimetric models as it satisfies triangle inequality, which has been proved in prior works (Pitis et al., 2020; Wang & Isola, 2022a; Liu et al., 2023):

$$\forall s_1, s_2, s_3, \ V^*(s_1; s_2) + V^*(s_2; s_3) \leq V^*(s_1; s_3). \quad (7)$$

Equation (7) shows that $V^* \in \mathcal{Q}^-(\mathcal{S})$. Therefore, we can use $-d_\theta$ to fit goal-conditioned value functions.

## 3. Motivation: Value Overestimation in Cross-trajectory Sampling

In this section, we consider value learning with a static dataset $\mathcal{D}$, where sampled state-goal pairs fall into the following two categories:

**Definition 3.1.** Given a state set $\mathcal{S}$, an action set $\mathcal{A}$, and a dataset $\mathcal{D} = \{\tau_i \mid i \in \{1, \ldots, n\}\}$, the state-goal pair $(s, g) \in \mathcal{S} \times \mathcal{S}$ is **in-trajectory** if there exists a trajectory $\tau = \{s_0, a_0, s_1, \ldots, s_{T-1}, a_{T-1}, s_T\}$ in the dataset and indices $i, j \in \{0, 1, \ldots, T\}$ such that: $s_i = s$, $s_j = g$, $i < j$. Otherwise, the state-goal pair $(s, g)$ is called **cross-trajectory**.

As shown in Equation (4), when $\alpha = 1$, only in-trajectory state-goal pairs are sampled, akin to HER. As $\alpha$ decreases, arbitrary states are more likely to be sampled as targets, leading to a higher probability of selecting cross-trajectory pairs. To demonstrate the challenge of value function learning for cross-trajectory pairs, we conduct experiments to evaluate the performance under different values of the parameter $\alpha$. As shown in Figure 1, HER performs well on goal-navigating tasks (left), where pairs of initial states and goals are in-trajectory, but fails on goal-stitching tasks involving cross-trajectory pairs (right). Moreover, incorporating cross-trajectory pairs into value learning with a lower $\alpha$ degrades performance on both tasks. To further clarify this issue, we first provide a definition to categorize cross-trajectory pairs into the following two types:

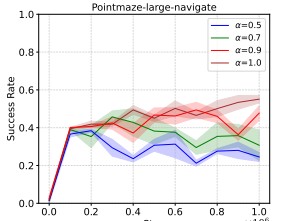 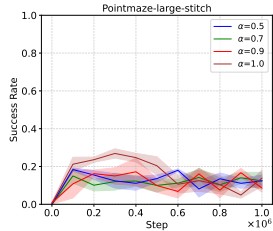

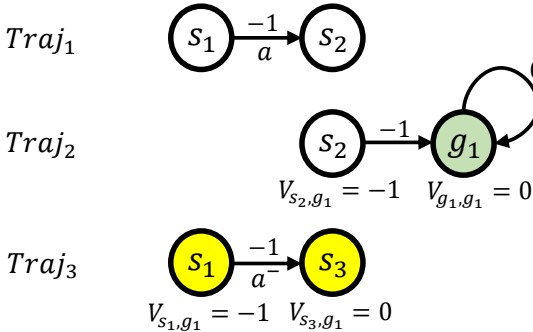

*Figure 1.* Performance of the GCIVL, on two different tasks under varying $\alpha$. In PointMaze-large-navigate (left), pairs of initial states and goals are in-trajectory in the dataset. The algorithm achieves optimal performance at $\alpha = 1.0$. In PointMaze-large-stitch (right), pairs of initial states and goals are cross-trajectory in the dataset. The policy fails to complete this task across all $\alpha$ settings.

*Figure 2.* Illustration of an unconnected state-goal pair in offline GCRL. The reward function is defined as $-1$ for each transition and $0$ upon reaching the goal and the discount factor is set to $\gamma = 1$. Solid black lines represent transitions present in the dataset. The target goal $g_1$ is represented by the green circle. The state-goal pair $(s_3, g_1)$ is unconnected, while other state-goal pairs are connected. The values of the $s_1$ and $s_2$ can be overestimated via the optimal Bellman operator, as represented by the yellow circle.

**Definition 3.2.** Given a state set $\mathcal{S}$, an action set $\mathcal{A}$, the static dataset can be represented as $\mathcal{D} = \{(s_i, a_i, s_{i+1}) \mid s_i, s_{i+1} \in \mathcal{S}, a_i \in \mathcal{A}\}$. A state-goal pair $(s, g) \in \mathcal{S} \times \mathcal{S}$ is **connected** if there exists a finite sequence $\{s_0, a_0, s_1, \ldots, s_{n-1}, a_{n-1}, s_n\}$ such that: $s_0 = s$, $s_n = g$, $(s_k, a_k, s_{k+1}) \in \mathcal{D}$ for all $k \in \{0, 1, \ldots, n-1\}$. Otherwise, the state-goal pair $(s, g)$ is called **unconnected**.

The unconnected state-goal pairs, whose values cannot be estimated accurately, may induce the policy to incorrect actions. Furthermore, the estimation error can be propagated to other states through TD learning. We consider the example illustrated in Figure 2 to provide an easy-to-understand explanation. Assume the discount factor $\gamma = 1$ and the reward function is $-1$ for every transition, except when the goal is reached, where the reward is $0$. The value function is updated via the optimal Bellman operator. According to Definition 3.2, the state-goal pair $(s_3, g_1)$ is unconnected, with its value $V(s_3, g_1)$ assumed to be randomly initialized to $0$. As there is no valid path in the dataset $\mathcal{D}$ that connects $s_3$ to $g_1$, the value $V(s_3, g_1)$ cannot be updated and could remain its initial overestimated value during TD learning. Then, $V(s_3, g_1)$ remains fixed at its initial overestimated value. This overestimation affects policy extraction at $s_1$, where the policy $\pi(s_1, g_1)$ incorrectly favors action $a^-$, which leads to $s_3$, over the correct action $a$, since $V(s_2, g_1) = -2 < V(s_3, g_1) = -1$. Furthermore, based on the optimal Bellman iteration, the connected state-goal pair $(s_1, g_1)$ is incorrectly estimated to $-1$ instead of the expected value $-2$ according to the valid trajectory $(s_1 \to s_2 \to g_1)$.

Let $\zeta_k(s, g) = |V_k(s, g) - V^*(s, g)|$ denote the total error at iteration $k$ of V-learning, and let $\delta_k(s, g) = |V_k(s, g) - \mathcal{T}V_{k-1}(s, g)|$ denote the Bellman iteration error (Kumar et al., 2019), where $\mathcal{T}$ is the optimal Bellman operator. Then, we derive the following bound on the total error at iteration $k$:

**Theorem 3.3** (Error Propagation in Goal-Conditioned Value Learning).

$$\zeta_k(s, g) \leq \delta_k(s, g) + \gamma \max_{a, s' \sim T} \zeta_{k-1}(s', g).$$

*Proof.* See proof in Appendix B.2. □

For any unconnected state-goal pair $(s', g)$, its error $\zeta_{k-1}(s', g)$ tends to be large due to the difficulty in estimating its true value from the dataset $\mathcal{D}$. Consequently, for connected state-goal pair $(s, g)$ adjacent to $(s', g)$ (i.e., there exists an action $a$ such that $(s, a, s') \in \mathcal{D}$), Theorem 3.3 implies that the error $\zeta_k(s, g)$ may also be large, as it inherits the propagated error from $\zeta_{k-1}(s', g)$.

To mitigate this issue, it is desirable to prevent the selection of actions $a$ that lead to unconnected state-goal pairs $(s', g)$ during the iterative process. This requires introducing additional mechanisms to underestimate values for unconnected state-goal pairs, which we describe in the next section.

## 4. Conservative Goal-Conditioned Implicit V-Learning

In this section, we develop a novel algorithm, Conservative Goal-Conditioned Implicit V-learning (GCIVL), to avoid the overestimation of value on unconnected state-goal pairs in offline settings. The core idea is to penalize the value of unconnected pairs while ensuring accurate value estimation for connected pairs through a quasimetric framework as discussed in Section 4.1 and Section 4.2. We detail the practical

implementation of the proposed algorithm and highlights key techniques that ensure its efficiency in Section 4.3.

## 4.1. Conservative Goal-Conditioned Off-Policy Evaluation

We aim to estimate the value function $V^\pi(s, g)$ for a target policy $\pi$ given access to a dataset $\mathcal{D}$ generated by a behavior policy $\pi_\beta$. In order to prevent overestimation of the value function for unconnected state-goal pairs, we introduce a penalty term that minimizes the value of state-goal pairs sampled from a specific distribution $p_\mu(s, g)$. Since the value function training process does not query the value function at unobserved states, we restrict $p_\mu$ to match the marginal state distribution in the dataset, such that $p_\mu(s, g) = d^{\pi_\beta}(s)\mu(g|s)$, where $\mu(g|s)$ denotes an arbitrary distribution which satisfies $\operatorname{supp}\mu \subset \operatorname{supp} p_m^\alpha$, $d^{\pi_\beta}(s)$ is the state marginal distribution of the dataset and is related to $\pi_\beta$. The resulting update formula, as a function of a tradeoff factor $\eta$, is as follows:

$$
\hat{V}_{k+1}^\pi \leftarrow \min_V \Big( \eta \mathbb{E}_{\substack{s \sim d^{\pi_\beta}(s), \\ g \sim \mu(g|s)}}[V(s,g)]
$$
$$
+ \mathbb{E}_{\substack{s \sim d^{\pi_\beta}(s), \\ g \sim p_m^\alpha(g|s)}} \Big( \hat{\mathcal{B}}^\pi \hat{V}_k(s,g) - V(s,g) \Big)^2 \Big),
$$
(8)

where $\hat{V}$ represents an empirical estimate of the true value function $V$, $\hat{B}$ denotes the empirical Bellman operator, which is the sample-based counterpart of the theoretical Bellman operator. In Theorem 4.1, we demonstrate that the learned value function $\hat{V}^\pi$, defined as $\hat{V}^\pi := \lim_{k\to\infty} \hat{V}_k^\pi$, serves as a conservative lower bound for the $V^\pi$ at all state-goal pairs.

**Theorem 4.1.** *For any $\mu(g|s)$ with $\operatorname{supp}\mu \subset \operatorname{supp} p_m^\alpha$ and $\delta \in (0,1)$, with probability $\geq 1 - \delta$, the V-function $\hat{V}^\pi$ obtained by iterating Equation (8) satisfies:*

$$
\forall s \in \mathcal{D}, g, \hat{V}^\pi(s,g) \leq V^\pi(s,g) - \eta \left[ (I - \gamma P^\pi)^{-1} \frac{\mu}{p_m^\alpha} \right]
$$
$$
(s,g) + \left[ (I - \gamma P^\pi)^{-1} \frac{C_{r,T,\delta} R_{\max}}{(1-\gamma)\sqrt{|\mathcal{D}|}} \right] (s,g).
$$

*Proof.* See proof in Appendix B.3. □

Given Theorem 4.1, it follows that the value $\hat{V}^\pi$ is underestimated across all state-goal pairs with an appropriate choice of parameter $\eta$. However, it is important to avoid underestimation on values of in-trajectory state-goal pairs. In practice, we replace $\mu(g|s)$ with a uniform random distribution $p_{\text{rand}}^{\mathcal{D}}(g)$, which equally samples across all goals in the dataset. Assuming that the empirical Bellman operator

is unbiased, the estimated value function is given by:

$$
\hat{V}^\pi(s,g) = V^\pi(s,g) - \eta \left[ (I - \gamma P^\pi)^{-1} \frac{p_{\text{uni}}^{\mathcal{D}}}{p_m^\alpha} \right] (s,g). \quad (9)
$$

We then provide a formal proof showing that, under suitable conditions, the value function accurately estimates in-trajectory state-goal pairs while preserving conservative estimates on cross-trajectory pairs.

**Proposition 4.2.** *For any $\epsilon > 0$, there exists a penalty factor $\eta$ such that the learned value function in Equation (9) satisfies:*

$$
\hat{V}^\pi(s^+, g) \geq V^\pi(s^+, g) - \epsilon,
$$

*where $(s^+, g)$ denotes any in-trajectory state-goal pairs in the dataset $\mathcal{D}$.*

*Proof.* See proof in Appendix B.4. □

Proposition 4.2 demonstrates that in-trajectory state-goal pairs can be accurately estimated. Subsequently, it is shown that cross-trajectory pairs are assigned substantially lower value estimates compared to in-trajectory pairs.

**Proposition 4.3.** *For any $\epsilon > 0$ with a static $\eta$, $\eta > 0$, there exists an $\alpha$ such that the learned value function satisfies:*

$$
\hat{V}^\pi(s^-, g) < \hat{V}^\pi(s^+, g) - \epsilon,
$$

*where $(s^+, g)$ denotes any in-trajectory state-goal pairs, and $(s^-, g)$ denotes any cross-trajectory state-goal pairs.*

*Proof.* See proof in Appendix B.5 □

## 4.2. Quasimetric-Based Value Estimation for Cross-Trajectory State-Goal Pairs

As shown in Proposition 4.3, the values of cross-trajectory state-goal pairs are underestimated, encompassing all unconnected pairs. However, cross-trajectory pairs that are connected can also be underestimated. As a result, we leverage a quasimetric model to represent the value function so that the value of such pair can be bounded via triangle inequality. As illustrated in Figure 2, $s_1$ and $g_1$ are connected but belong to different trajectories with intermediate nodes $s_2$. Since $V(s_1, s_2)$ and $V(s_2, g_1)$ can be accurately estimated given Proposition 4.2, the value of $V(s_1, g_1)$ can be bounded by the triangle inequality.

**Proposition 4.4.** *Suppose the learned value function $\hat{V}^\pi$ satisfies the properties of quasimetric. For any $\epsilon > 0$, and the connected state-goal pair $(s, g)$ with intermediate states $s_1, s_2, \ldots, s_n$ observed in the dataset, there exists a penalty factor $\eta$ such that:*

$$
\hat{V}^\pi(s,g) \geq \sum_{i=0}^n V^\pi(s_i, s_{i+1}) - \epsilon,
$$

where $s_0 = s, s_{n+1} = g$.

*Proof.* The proof follows directly from the properties of the quasimetric and Proposition 4.2. The triangle inequality ensures that the accumulated values of sub-segments $(s, s_i)$ and $(s_i, g)$ provide a valid lower bound for the total value $(s, g)$. Adjusting for a small $\epsilon$ through appropriate tuning of $\eta$ completes the proof. $\square$

Proposition 4.4 guarantees that the value of any connected state-goal pairs are bounded by the sum of sub-segment values no matter whether they belong to the same trajectory. Building on this property, we further leverage the quasimetric, in conjunction with the penalty-based conservative value estimation, to achieve a clear separation between connected and unconnected state-goal pairs.

**Proposition 4.5.** *For any $\eta > 0$ and $\epsilon > 0$, there exists a hyperparameter $\alpha$ such that the learned value function $\hat{V}^\pi$ satisfies:*

$$\hat{V}^\pi(s^-, g) \leq \hat{V}^\pi(s^+, g) - \epsilon,$$

*where $(s^+, g)$ represents any connected state-goal pairs, and $(s^-, g)$ are any unconnected pairs.*

*Proof.* See proof in Appendix B.6. $\square$

By combining Proposition 4.4 and Proposition 4.5, we show that incorporating quasimetric constraints into value function estimation ensures reasonable lower bounds for connected state-goal pairs while maintaining underestimation for unconnected pairs.

### 4.3. A Practical Algorithm

Our algorithm is implemented based on Goal-Conditioned Implicit V-Learning (Park et al., 2023), which is a variant of Implicit Q-Learning (IQL) (Kostrikov et al., 2021) designed for handling goal-conditioned tasks in the offline setting. As stated in Appendix A, at any time step $k$, GCIVL can be interpreted as performing policy evaluation for a specific policy $\pi_k^\tau$. Given this observation, we incorporate a penalty term into the original loss function, leading to the following updated formula:

$$
\begin{aligned}
V_{k+1} = \min_V \mathbb{E}_{\substack{s \sim d^{\pi_\beta}(s), g \sim p_m^\alpha(g|s) \\ s' \sim p(s'|s,g)}} [\ell_\tau^2(r(s,g) + \gamma V_k(s',g) \\
- V(s,g)) + \eta \mathbb{E}_{\substack{s \sim d^{\pi_\beta}(s), \\ g \sim p_{\text{rand}}^{\mathcal{D}}(g)}} [V(s,g)]],
\end{aligned}
\tag{10}
$$

where $V \in \mathcal{Q}^-(\mathcal{S})$ ensures that the value function satisfies the property of quasimetric.

#### 4.3.1. QUASIMETRIC DISTILLATION

In goal-conditioned problems, the optimal value function actually obeys the triangle inequality. However, during TD learning, intermediate values may not naturally adhere to this property, so directly enforcing the quasimetric constraint can negatively impact learning performance (Wang et al., 2020; 2021). To address this issue, we use a dual-network approach: an unrestricted network $\theta_v$ for TD learning and a quasimetric-constrained network $\theta_d \in \mathcal{Q}^-(\mathcal{S})$. The expectile regression loss for $\theta_v$ is defined as:

$$
\begin{aligned}
\mathcal{L}_{\theta_v} = \mathbb{E}_{\substack{s \sim d^{\pi_\beta}, g \sim p_m^\alpha \\ s' \sim T(s'|s,g)}} [\ell_\tau^2(r(s,g) + \gamma V_k(s',g) \\
- V(s,g))].
\end{aligned}
\tag{11}
$$

Then, we distill values from $\theta_v$ into $\theta_d$ and add a penalty term to underestimate values of unconnected state-goal pairs. The loss for $\theta_d$ is defined as:

$$
\begin{aligned}
\mathcal{L}_{\theta_d} = \mathbb{E}_{\substack{s \sim d^{\pi_\beta}, g \sim p_m^\alpha \\ s' \sim T(s'|s,g)}} [\ell^2(V_{\theta_v}(s,g) - V_{\theta_d}(s,g))] \\
+ \eta \mathbb{E}_{\substack{s \sim d^{\pi_\beta}, \\ g' \sim p_{\text{rand}}^{\mathcal{D}}(g)}} [V_{\theta_d}(s,g')].
\end{aligned}
\tag{12}
$$

Finally, to align $\theta_v$ with the conservative estimates of $\theta_d$, we introduce a supervised term in the loss function of $\theta_v$. This ensures that $\theta_v$ incorporates the conservative estimated of $V_{\theta_d}$ on unconnected pairs while retaining flexibility during TD updates. The final loss function for $\theta_v$ is:

$$
\begin{aligned}
\mathcal{L}_{\theta_v} = \mathbb{E}_{\substack{s \sim d^{\pi_\beta}, g \sim p_m^\alpha \\ s' \sim T(s'|s,g)}} [\ell_\tau^2(r(s,g) + \gamma V_k(s',g) - V(s,g))] \\
+ \rho \mathbb{E}_{\substack{s \sim d^{\pi_\beta}, \\ g' \sim p_{\text{rand}}^{\mathcal{D}}(g)}} [\ell^2(V_{\theta_v}(s,g') - V_{\theta_d}(s,g'))],
\end{aligned}
\tag{13}
$$

where $\rho$ is a tunable hyperparameter. The proposed dual-network framework effectively decouples value iteration (via $\theta_v$) from quasimetric constraints (via $\theta_d$). Ablation studies in Section 5.3 demonstrate that this structure improves the efficiency of value learning.

#### 4.3.2. HIERARCHICAL FRAMEWORK

As highlighted in prior works, existing offline reinforcement learning algorithms often struggle with long-horizon tasks where the goal is far away, primarily due to the signal-to-noise ratio challenge (Park et al., 2023). An effective approach to addressing this issue is to leverage a hierarchical policy framework. Specifically, we extract both a high-level policy $\pi_{\theta_h}^h(s_{t+k} \mid s_t, g)$ and a low-level policy $\pi_{\theta_\ell}^\ell(a \mid s_t, s_{t+k})$, aiming to maximize $V(s_{t+k}, g)$ and $V(s_{t+1}, s_{t+k})$, respectively. Here, $s_{t+k}$ is treated as a waypoint or sub-goal, where $k$ represents the step size.

The high-level policy outputs sub-goals $s_{t+k}$ based on the current state $s_t$ and the final goal $g$. The low-level policy

**Algorithm 1** CGCIVL

**Input:** offline dataset $\mathcal{D}$

initialize unconstrained value function $V_{\theta_v}(s, g)$, quasi-metric value function model $V_{\theta_d}(s, g)$, high-level policy $\pi^h_{\theta_h}(s_{t+k}|s_t, g)$, low-level policy $\pi^\ell_{\theta_\ell}(a|s_t, s_{t+k})$

**while** not converged **do**

$\quad \theta_v \leftarrow \theta_v - \lambda_v \nabla_{\theta_v} \mathcal{L}_V(\theta_V)$ {# Equation (13)}

$\quad \theta_d \leftarrow \theta_d - \lambda_d \nabla_{\theta_d} \mathcal{L}_d(\theta_d)$ {# Equation (12)}

$\quad \theta_h \leftarrow \theta_h - \lambda_h \nabla_{\theta_h} \mathcal{L}_h(\theta_h)$ {# Equation (14)}

$\quad \theta_\ell \leftarrow \theta_\ell - \lambda_\ell \nabla_{\theta_\ell} \mathcal{L}_\ell(\theta_\ell)$ {# Equation (15)}

**end while**

then outputs appropriate actions $a$ conditioned on the current state and the sub-goal. The loss functions for the high-level and low-level policies are as follows:

$$\mathcal{L}_{\theta_h} = \mathbb{E}_{(s_t, s_{t+k}, g)}[- \exp(\beta \cdot \tilde{A}^h(s_t, s_{t+k}, g)) \\ \log \pi^h_{\theta_h}(s_{t+k} \mid s_t, g)], \tag{14}$$

$$\mathcal{L}_{\theta_\ell} = \mathbb{E}_{(s_t, a_t, s_{t+1}, s_{t+k})}[- \exp(\beta \cdot \tilde{A}^\ell(s_t, a_t, s_{t+k})) \\ \log \pi^\ell_{\theta_\ell}(a_t \mid s_t, s_{t+k})], \tag{15}$$

where we approximate $\tilde{A}^h(s_t, s_{t+k}, g)$ as $V_{\theta_v}(s_{t+k}, g) - V_{\theta_v}(s_t, g)$ and $\tilde{A}^\ell(s_t, a_t, s_{t+k})$ as $V_{\theta_v}(s_{t+1}, s_{t+k}) - V_{\theta_v}(s_t, s_{t+k})$. The pseudocode of CGCIVL is shown in Algorithm 1.

# 5. Experiments

We conduct a series of experiments to evaluate the effectiveness of the proposed algorithm in addressing offline goal-conditioned tasks. In Section 5.1, we describe the experimental setup, including the environments with different configurations and the dataset types used for training. In Section 5.2, we compare CGCIVL[1] with several strong baseline methods across diverse environments and datasets, demonstrating its superior performance in both goal-navigating and goal-stitching tasks. In Section 5.3, we analyze ablation studies to evaluate the sensitivity of the cross-trajectory goal-sampling ratio and the penalty coefficient. Furthermore, we discuss the contribution of the quasimetric distillation and the hierarchical policy structure to the overall performance.

## 5.1. Experimental Setup

We evaluate the proposed algorithm on **OGbench** (Park et al., 2024), a benchmark designed to evaluate algorithms in offline GCRL across different tasks and datasets. The experiments are conducted in three locomotion environments (PointMaze, AntMaze, HumanoidMaze) and three manipulation environments (Cube, Puzzle, Scene). In each task, the policy is evaluated across five different goals, with an

---

[1]Implementation details can be found in Appendix C.2

average success rate computed over 50 trials per goal. In our experiments, we report 95% confidence intervals as shaded regions in figures or standard deviations in tables, unless otherwise specified. We provide more details of environments and datasets in Appendix D.

Our algorithm is compared with the following approaches: 1) **Goal-Conditioned Behavioral Cloning (GCBC)** (Lynch et al., 2020; Ghosh et al., 2021), a straightforward goal-conditioned method that learns policies by mimicking the trajectories in the dataset; 2) **Goal-Conditioned Implicit V-Learning (GCIVL)** (Park et al., 2023), an offline GCRL algorithm that uses expectile regression to approximate the optimal value functions; 3) **Goal-Conditioned Implicit Q-Learning (GCIQL)** (Kostrikov et al., 2021), which, like GCIVL, applies expectile regression but focuses on approximating $Q$-values instead; 4) **Quasimetric Reinforcement Learning (QRL)** (Wang et al., 2023), a non-traditional GCRL method that fits a quasimetric value function with a dual-objective framework; 5) **Contrastive Reinforcement Learning (CRL)** (Eysenbach et al., 2022), a one-step RL algorithm that learns value functions through contrastive learning and performs single-step policy improvement; and 6) **Hierarchical Implicit Q-Learning (HIQL)** (Park et al., 2023), a hierarchical RL algorithm that derives a two-level policy based on GCIVL.

## 5.2. Performance on Offline Goal-Conditioned Tasks

The results in Table 1 demonstrate that CGCIVL consistently outperforms or matches all baseline algorithms across most tasks in every environment. In locomotion environments, our algorithm achieves state-of-the-art performance and particularly excels in long-horizon and goal-stitching tasks. For instance, on PointMaze-giant-stitch, CGCIVL achieves an 81% success rate, surpassing the second-best method (50%) by a substantial margin. Similarly, on both AntMaze-giant-stitch and HumanoidMaze-giant-stitch, CGCIVL exhibits a clear advantage. In addition, in goal-navigating tasks, the proposed algorithm also demonstrates improvements over prior methods. This advantage arises from the fact that trajectories in the dataset are suboptimal, and CGCIVL effectively leverages the strengths of different trajectories to construct more efficient goal-reaching strategies. Moreover, our algorithm also achieves nearly optimal results in the manipulation environment, particularly in the Cube and Scene. This indicates that our proposed algorithm is capable of learning diverse trajectory-based skills to accomplish more complex goals.

The significant effectiveness of the proposed algorithm stems from its more accurate value function estimation. As shown in Figure 3, in GCIVL, the value function shows little variation for states far from the goal, remaining around $-200$. Specifically, with the goal located in the bottom-left,

values for states in the bottom-right corner, despite being closer to the goal, are noticeably similar to those in the top-right corner. In contrast, CGCIVL achieves more accurate value estimates, especially for states far from the goal.

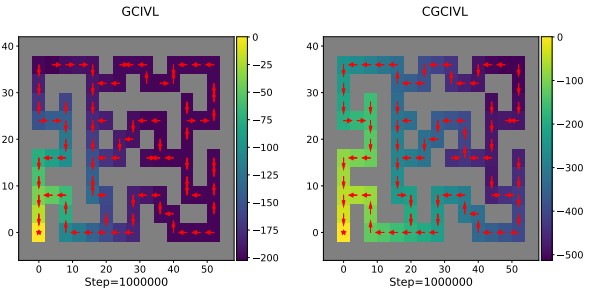

*Figure 3.* Estimated value functions of GCIVL and CGCIVL in the PointMaze-giant-stitch environment ($\alpha = 0.7$). The goal location is marked with a red star. The magnitude of the value function is represented by the color scale, with warmer colors indicating higher values. Arrows depict the actions chosen by the policy.

### 5.3. Ablation Study

**Ablation 1**    As shown in Figure 4, we analyze the performance of CGCIVL under the varying cross-trajectory goal-sampling ratio, controlled by the parameter $\alpha$. In goal-navigating tasks, increasing $\alpha$ achieves higher success rates by encouraging the algorithm to focus more on in-trajectory goals during training. However, as $\alpha$ reaches higher values (e.g., 0.7, 0.9, 1.0), the algorithm's performance remains relatively stable, largely due to its accurate value function estimation, even with a certain probability of random goal sampling. In contrast, in goal-stitching tasks, reducing $\alpha$ brings significant improvements in experimental results, as sampled state-goal pairs in the test often span across multiple trajectories in the dataset, requiring the agent to prioritize cross-trajectory goals during the training process.

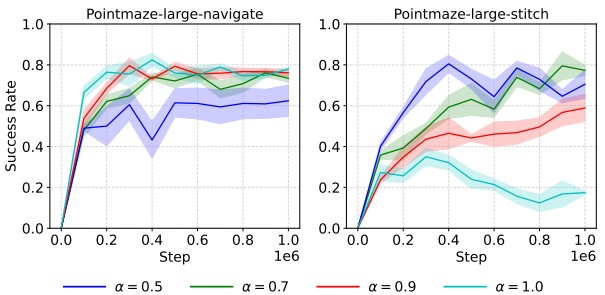

*Figure 4.* Performance of CGCIVL on two different environments under varying $\alpha$. The success rate is shown across different training steps, with shaded regions representing the standard deviation.

**Ablation 2**    As shown in Figure 5, we evaluate the performance of CGCIVL under different penalty coefficients $\eta$. The results indicate that both excessively small and large $\eta$ degrade the performance. Specifically, when $\eta = 0.0$, the conservative regularization is entirely removed, and the algorithm struggles on both navigation and stitching tasks. This suggests that the algorithm fails to learn an accurate value function due to the overestimation of values on unconnected state-goal pairs. Conversely, an excessively large $\eta$ (e.g. $\eta = 1.0$) also harms performance, as it over-constrains the optimization and limits the ability of the model to fit the true value effectively. In practice, we determine the optimal $\eta$ through empirical validation across multiple candidate values.

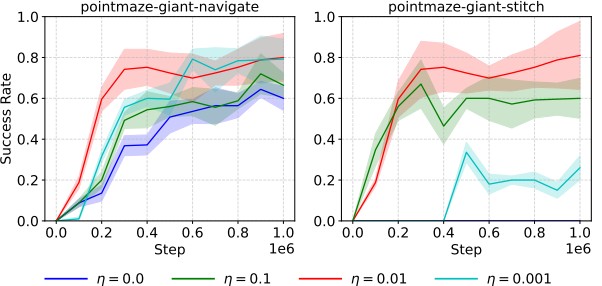

*Figure 5.* Performance of CGCIVL on two different environments under varying $\eta$. The success rate is shown across different training steps, with shaded regions representing the standard deviation.

**Ablation 3**    The results in Figure 6 highlight the significant contributions of both the hierarchical policy framework and the quasimetric distillation mechanism to CGCIVL's performance. The figure shows that hierarchical architectures excel in long-horizon tasks through sub-goal selection and long-term planning but offer limited benefits in simpler environments. In addition, the quasimetric distillation mechanism is crucial for accurate value estimation, especially for cross-trajectory goals. Its removal severely impacts success rates, with the most pronounced effects observed in stitching tasks. These findings emphasize that the hierarchical policy framework enables efficient planning, while the quasimetric distillation mechanism ensures robust value estimation for goal stitching and long-horizon reasoning.

## 6. Related Work

**Goal-Conditioned RL**    GCRL focuses on learning a universal value function  (Schaul et al., 2015), enabling agents to achieve various goals. Prior work on goal-conditioned RL has explored a diverse range of techniques, such as contrastive learning (Eysenbach et al., 2021; 2022), state-occupancy matching (Durugkar et al., 2021; Ma et al., 2022a) and successor features (Borsa et al., 2018; Touati

*Table 1.* Experimental results for PointMaze, AntMaze, HumanoidMaze, Cube, Scene and Puzzle across diverse datasetsets. The table reports the average binary success rate (%) across five test-time goals for each task, averaged over 8 seeds. Standard deviations are indicated by the $\pm$ symbol. Entries within 95% of the best-performing value in each row are highlighted in **bold**.

| Environment | Dataset Type | Dataset | GCBC | GCIVL | GCIQL | QRL | CRL | HIQL | CGCIVL |
|---|---|---|---|---|---|---|---|---|---|
| PointMaze | navigate | pointmaze-medium-navigate-v0 | $9 \pm 6$ | $63 \pm 6$ | $53 \pm 8$ | $82 \pm 5$ | $29 \pm 7$ | $79 \pm 15$ | $\mathbf{87 \pm 4}$ |
| | | pointmaze-large-navigate-v0 | $29 \pm 6$ | $45 \pm 5$ | $34 \pm 3$ | $86 \pm 7$ | $39 \pm 7$ | $58 \pm 15$ | $\mathbf{92 \pm 4}$ |
| | | pointmaze-giant-navigate-v0 | $1 \pm 2$ | $0 \pm 0$ | $0 \pm 0$ | $68 \pm 7$ | $27 \pm 10$ | $46 \pm 9$ | $\mathbf{80 \pm 12}$ |
| | | **average performance** | 13.00 | 36.00 | 29.00 | 78.67 | 31.67 | 61.00 | **86.33** |
| | stitch | pointmaze-medium-stitch-v0 | $23 \pm 18$ | $70 \pm 14$ | $21 \pm 9$ | $80 \pm 12$ | $10 \pm 8$ | $74 \pm 16$ | $\mathbf{89 \pm 8}$ |
| | | pointmaze-large-stitch-v0 | $7 \pm 5$ | $12 \pm 6$ | $31 \pm 2$ | $84 \pm 15$ | $0 \pm 0$ | $13 \pm 6$ | $\mathbf{98 \pm 2}$ |
| | | pointmaze-giant-stitch-v0 | $0 \pm 0$ | $0 \pm 0$ | $0 \pm 0$ | $50 \pm 8$ | $0 \pm 0$ | $0 \pm 0$ | $\mathbf{81 \pm 17}$ |
| | | **average performance** | 10.00 | 27.33 | 17.33 | 71.33 | 3.33 | 29.00 | **89.33** |
| AntMaze | navigate | antmaze-medium-navigate-v0 | $29 \pm 4$ | $72 \pm 8$ | $71 \pm 4$ | $88 \pm 3$ | $\mathbf{95 \pm 1}$ | $\mathbf{96 \pm 1}$ | $\mathbf{95 \pm 1}$ |
| | | antmaze-large-navigate-v0 | $24 \pm 2$ | $16 \pm 5$ | $34 \pm 4$ | $75 \pm 6$ | $83 \pm 4$ | $\mathbf{91 \pm 2}$ | $\mathbf{91 \pm 2}$ |
| | | antmaze-giant-navigate-v0 | $0 \pm 0$ | $0 \pm 0$ | $0 \pm 0$ | $16 \pm 3$ | $16 \pm 3$ | $65 \pm 5$ | $\mathbf{73 \pm 5}$ |
| | | **average performance** | 17.67 | 29.33 | 35.00 | 59.67 | 64.67 | 84.00 | **86.33** |
| | stitch | antmaze-medium-stitch-v0 | $45 \pm 11$ | $44 \pm 6$ | $29 \pm 6$ | $59 \pm 7$ | $53 \pm 6$ | $\mathbf{94 \pm 1}$ | $91 \pm 3$ |
| | | antmaze-large-stitch-v0 | $3 \pm 3$ | $18 \pm 2$ | $7 \pm 2$ | $18 \pm 2$ | $11 \pm 2$ | $67 \pm 5$ | $\mathbf{79 \pm 3}$ |
| | | antmaze-giant-stitch-v0 | $0 \pm 0$ | $0 \pm 0$ | $0 \pm 0$ | $0 \pm 0$ | $0 \pm 0$ | $25 \pm 7$ | $\mathbf{36 \pm 7}$ |
| | | **average performance** | 16.00 | 20.67 | 12.00 | 25.67 | 21.33 | 62.00 | **68.67** |
| HumanoidMaze | navigate | humanoidmaze-medium-navigate-v0 | $8 \pm 2$ | $24 \pm 2$ | $27 \pm 2$ | $21 \pm 8$ | $60 \pm 4$ | $89 \pm 2$ | $\mathbf{91 \pm 3}$ |
| | | humanoidmaze-large-navigate-v0 | $1 \pm 0$ | $2 \pm 1$ | $2 \pm 1$ | $5 \pm 1$ | $24 \pm 4$ | $49 \pm 4$ | $\mathbf{58 \pm 8}$ |
| | | humanoidmaze-giant-navigate-v0 | $0 \pm 0$ | $0 \pm 0$ | $0 \pm 0$ | $1 \pm 0$ | $3 \pm 2$ | $24 \pm 7$ | $\mathbf{29 \pm 9}$ |
| | | **average performance** | 3.00 | 8.67 | 9.67 | 9.00 | 29.00 | 54.00 | **59.33** |
| | stitch | humanoidmaze-medium-stitch-v0 | $29 \pm 5$ | $12 \pm 2$ | $12 \pm 3$ | $18 \pm 2$ | $36 \pm 2$ | $88 \pm 2$ | $\mathbf{90 \pm 2}$ |
| | | humanoidmaze-large-stitch-v0 | $6 \pm 3$ | $1 \pm 1$ | $0 \pm 0$ | $3 \pm 1$ | $4 \pm 1$ | $28 \pm 3$ | $\mathbf{32 \pm 4}$ |
| | | humanoidmaze-giant-stitch-v0 | $0 \pm 0$ | $0 \pm 0$ | $0 \pm 0$ | $0 \pm 0$ | $0 \pm 0$ | $6 \pm 5$ | $\mathbf{34 \pm 6}$ |
| | | **average performance** | 11.67 | 4.33 | 4.00 | 7.00 | 13.33 | 40.67 | **52.00** |
| Cube | play | cube-single-play-v0 | $6 \pm 2$ | $53 \pm 4$ | $68 \pm 6$ | $5 \pm 1$ | $19 \pm 2$ | $15 \pm 3$ | $\mathbf{84 \pm 4}$ |
| | | cube-double-play-v0 | $1 \pm 1$ | $36 \pm 3$ | $40 \pm 5$ | $1 \pm 0$ | $10 \pm 2$ | $6 \pm 2$ | $\mathbf{46 \pm 4}$ |
| | | cube-triple-play-v0 | $1 \pm 1$ | $1 \pm 0$ | $3 \pm 1$ | $0 \pm 0$ | $4 \pm 1$ | $3 \pm 1$ | $\mathbf{5 \pm 2}$ |
| | | cube-quadruple-play-v0 | $\mathbf{0 \pm 0}$ | $\mathbf{0 \pm 0}$ | $\mathbf{0 \pm 0}$ | $\mathbf{0 \pm 0}$ | $\mathbf{0 \pm 0}$ | $\mathbf{0 \pm 0}$ | $\mathbf{0 \pm 0}$ |
| | | **average performance** | 2.00 | 22.50 | 27.75 | 1.50 | 8.25 | 6.00 | **33.75** |
| Scene | play | scene-play-v0 | $5 \pm 1$ | $42 \pm 4$ | $51 \pm 4$ | $5 \pm 1$ | $19 \pm 2$ | $38 \pm 3$ | $\mathbf{77 \pm 5}$ |
| Puzzle | play | puzzle-3x3-play-v0 | $2 \pm 0$ | $6 \pm 1$ | $\mathbf{95 \pm 1}$ | $1 \pm 0$ | $3 \pm 1$ | $12 \pm 2$ | $13 \pm 3$ |
| | | puzzle-4x4-play-v0 | $0 \pm 0$ | $13 \pm 2$ | $26 \pm 3$ | $0 \pm 0$ | $0 \pm 0$ | $0 \pm 0$ | $\mathbf{32 \pm 5}$ |
| | | puzzle-4x5-play-v0 | $0 \pm 0$ | $7 \pm 1$ | $\mathbf{14 \pm 1}$ | $0 \pm 0$ | $1 \pm 0$ | $4 \pm 1$ | $15 \pm 3$ |
| | | puzzle-4x6-play-v0 | $0 \pm 0$ | $10 \pm 1$ | $\mathbf{12 \pm 1}$ | $0 \pm 0$ | $4 \pm 1$ | $3 \pm 1$ | $9 \pm 2$ |
| | | **average performance** | 0.05 | 9.00 | **36.75** | 0.25 | 2.00 | 4.75 | 17.25 |

& Ollivier, 2021; Ghosh et al., 2023). One of the primary challenges in GCRL is the low learning efficiency caused by sparse rewards. HER (Andrychowicz et al., 2017) is proposed to address this issue, which treats the intermediate states in trajectories as new goals, thereby providing additional learning signals to the agent. Other works have explored alternative approaches, such as hierarchical frameworks (Kulkarni et al., 2016; Nachum et al., 2018; Chane-Sane et al., 2021), which decompose tasks into manageable subgoals, and model-based planning (Zhu et al., 2021; Mendonca et al., 2021), which utilizes predictive models to guide agents toward their goals.

In addition, several studies have discovered that the optimal value function in GCRL satisfies the triangle inequality (Pitis et al., 2020; Wang & Isola, 2022a; Liu et al., 2023), offering a novel perspective that the optimal value function can be interpreted as a generalized metric function, known

as a quasimetric. Based on this insight, various network architectures have been proposed to effectively represent quasimetric functions (Wang & Isola, 2022b;a; Liu et al., 2023). Utilizing these architectures to represent the value function helps constrain the function space, thereby accelerating value function convergence (Wang et al., 2023).

**Offline GCRL** Offline GCRL aims to learn goal-reaching policies from pre-collected datasets. Due to the lack of interactions with the environment, most GCRL algorithms struggle to perform well in offline settings. Some approaches adopt imitation learning methods, directly fitting policies to the trajectories in the dataset (Ghosh et al., 2021; Yang et al., 2022). However, these methods heavily depend on the quality of the collected trajectories. Other approaches adapt traditional offline reinforcement learning algorithms (Kumar et al., 2020; Kostrikov et al., 2021) to goal-conditioned

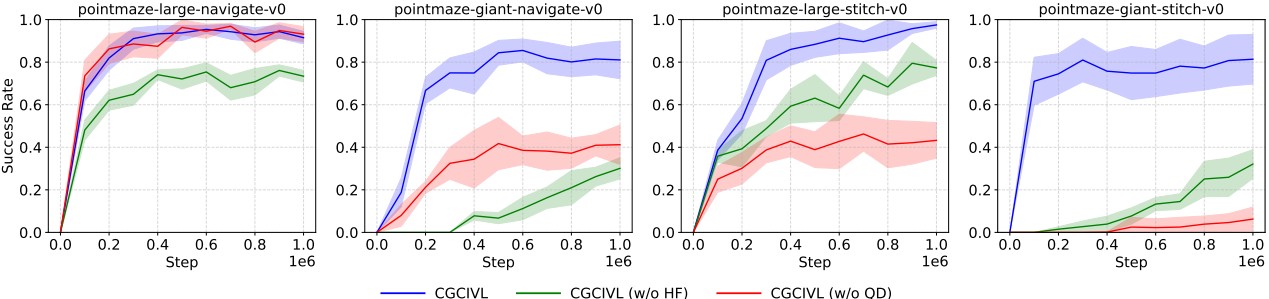

*Figure 6.* Ablation study results on four tasks: *pointmaze-large-navigate-v0*, *pointmaze-giant-navigate-v0*, *pointmaze-large-stitch-v0*, and *pointmaze-giant-stitch-v0*. Each plot compares the full CGCIVL algorithm against its ablated versions. **CGCIVL** (blue) represents the complete algorithm. **CGCIVL (w/o HF)** (green) removes the hierarchical policy framework, and **CGCIVL (w/o QD)** (red) removes the quasimetric distillation mechanism.

scenarios, avoiding the issue of distribution shift (Park et al., 2023). A key challenge in offline GCRL is goal stitching (Park et al., 2024). Previous works have utilized generative models to stitch different trajectories in the dataset (Kim et al., 2024; Li et al., 2024). However, these methods depend on the quality of the trained generative model, making the reliability of the generated trajectories difficult to ensure. In contrast, we focus on enabling the agent to learn and integrate knowledge from existing trajectories without augmenting the dataset.

## 7. Conclusion

In this work, we classify state-goal pairs in offline settings and identify a key challenge in value estimation for cross-trajectory pairs—some of these pairs may be unconnected. Values of those unconnected state-goal pairs cannot be estimated accurately, and errors may be propagated to other states, thus misleading policy to extract suboptimal actions. To address this problem, we propose a novel algorithm that incorporates a regularization term and quasimetric constraints during the value function learning process. This approach ensures the underestimation of unconnected state-goal pairs while providing accurate value estimates for connected pairs. Experiments across multiple environments and datasets demonstrate that our method significantly improves performance on goal-conditioned tasks, especially in goal-stitching scenarios.

Future research could focus on developing a more robust and generalizable framework for learning reduced goal spaces from offline datasets, thereby improving the algorithm's scalability and adaptability to diverse and complex environments. Additionally, an interesting direction could be exploring strategies to identify more suitable distributions for underestimating the value function of state-goal pairs, thereby improving the stability of the learning process.

## Acknowledgments

We gratefully acknowledge the support from the Distinguished Young Scholars Project funded by the Natural Science Foundation of Guangdong Province (No. 2025B1515020060), the Basic and Applied Basic Research Program of the Guangzhou Science and Technology Plan (No. 2025A04J7141), and the Xiaomi Young Talents Program.

## Impact Statement

This paper presents work whose goal is to advance the field of Machine Learning. There are many potential societal consequences of our work, none of which we feel must be specifically highlighted here.

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

## A. Analysis of GCIVL

GCIVL is a variant of Implicit Q-Learning (IQL) (Kostrikov et al., 2021) designed for handling goal-conditioned tasks in the offline setting. GCIVL leverages expectile regression (Newey & Powell, 1987) to learn the goal-conditioned value function, updated iteratively as follows:

$$V_{k+1} = \min_V \mathbb{E}_{\substack{s \sim \mathcal{D}, g \sim p_m^\alpha(g|s), \\ s' \sim p(s'|s,g)}} [\ell_\tau^2(r(s,g) + \gamma V_k(s',g) - V(s,g))], \tag{16}$$

where $\ell_\tau^2(x) = |\tau - \mathbf{1}_{\{x<0\}}(x)| x^2$ denotes the expectile loss with an expectile $\tau \in [0.5, 1)$.

**Lemma A.1.** *In the deterministic environment described above, there exists a policy $\pi_k^\tau$ such that the value function update in Equation* (16) *can be equivalently expressed as:*

$$V_{k+1} = \arg\min_V \mathbb{E}_{\substack{s \sim \mathcal{D}_\mathcal{S}, g \sim p_m^\alpha(g|s) \\ a \sim \pi_k^\tau, s' \sim T(\cdot|s,a)}} [(r(s,g) + \gamma V_k(s',g) - V(s,g))^2], \tag{17}$$

*where $T(\cdot \mid s,a)$ represents the deterministic transition dynamics, and $\pi_k^\tau$ is a policy that depends on the current value function $V_k$.*

Based on the above lemma, GCIVL can be interpreted as a composite process of value function evaluation and policy improvement. Specifically, $V_{k+1}$ is computed via policy evaluation by iterating the Bellman operator:

$$\mathcal{B}^{\pi_k^\tau} V_k = r + \gamma P^{\pi_k^\tau} V_k, \tag{18}$$

where the transition operator $P^{\pi_k^\tau} V_k$ is defined as:

$$P^{\pi_k^\tau} V_k(s,g) = \mathbb{E}_{a \sim \pi_k^\tau(\cdot|s,g), s' \sim T(\cdot|s,a)}[V_k(s',g)]. \tag{19}$$

As $\tau \to 1$, the operator converges to the optimal Bellman operator $\mathcal{T}$ (Ghosh et al., 2023).

## B. Proofs

### B.1. Proof of Lemma A.1

*Proof.*

$$V_{k+1} = \min_V \mathbb{E}_{\substack{s \sim \mathcal{D}_\mathcal{S}, \\ g \sim p_m^\alpha(g|s)}} [\ell_\tau^2(r(s,g) + \gamma V_k(s',g) - V(s,g))] \tag{20}$$

$$= \min_V \mathbb{E}_{p_\mathcal{D}(s) p_m^\alpha(g|s) p_{\pi_\beta}(s'|s,g)} [\ell_\tau^2(r(s,g) + \gamma V_k(s',g) - V(s,g))], \tag{21}$$

where $p_\mathcal{D}(s)$ denotes uniformly sampling state $s$ from the dataset, and $p_{\pi_\beta}(s'|s,g)$ represents the probability of $s'$ being adjacent to $s$ in the dataset, conditioned on $s$ and goal $g$. We assume:

$$U_s = \{s' \mid p_{\pi_\beta}(s' \mid s,g) > 0, s' \in \mathcal{D}\}, \tag{22}$$

$$U_s^+ = \{s' \mid r(s,g) + \gamma V_k(s',g) > V(s,g), s' \in U_s\}, \tag{23}$$

$$U_s^- = U_s - U_s^+. \tag{24}$$

Thus:

$$V_{k+1} = \min_V \mathbb{E}_{p_\mathcal{D}(s) p_m^\alpha(g|s)} \Big[ \sum_{s' \in U_s^+} \tau (r(s,g) + \gamma V_k(s'g) - V(s,g))^2 \cdot p_{\pi_\beta}(s' \mid s,g)$$
$$+ \sum_{s' \in U_s^-} |1 - \tau| \cdot (r(s,g) + \gamma V_k(s'g) - V(s,g))^2 \cdot p_{\pi_\beta}(s' \mid s,g) \Big]. \tag{25}$$

By setting the derivative of Equation (25) to 0, we obtain the following expression for $V_{k+1}$:

$$V_{k+1}(s,g) = \frac{\sum_{s' \in U_s^+} \tau p_{\pi_\beta}(s'|s,g) \cdot \mathcal{B}_k(s,g,s') + \sum_{s' \in U_s^-} \tau p_{\pi_\beta}(s'|s,g) \cdot \mathcal{B}_k(s,g,s')}{\tau \sum_{s' \in U_s^+} p_{\pi_\beta}(s'|s,g) + |1 - \tau| \cdot \sum_{s' \in U_s^-} p_{\pi_\beta}(s'|s,g)}, \tag{26}$$

where $\mathcal{B}_k(s, g, s') = r(s, g) + \gamma V_k(s', g)$. We denote:

$$\tilde{p}(s' \mid s, g) = \begin{cases} \frac{\tau \cdot p_{\pi_\beta}(s' \mid s, g)}{\tau \sum_{s' \in U_s^+} p_{\pi_\beta}(s' \mid s, g) + (1-\tau) \sum_{s' \in U_s^-} p_{\pi_\beta}(s' \mid s, g)}, & s' \in U_s^+ \\ \frac{(1-\tau) \cdot p_{\pi_\beta}(s' \mid s, g)}{\tau \sum_{s' \in U_s^+} p_{\pi_\beta}(s' \mid s, g) + (1-\tau) \sum_{s' \in U_s^-} p_{\pi_\beta}(s' \mid s, g)}, & s' \in U_s^- \end{cases}, \tag{27}$$

where $\sum_{s' \in U_s} \tilde{p}(s' \mid s, g) = 1$. We could express $V_{k+1}(s, g) = \sum_{s' \in U_s} \tilde{p}(s' \mid s, g) \cdot \mathcal{B}_k(s, g, s')$. Since the environment is deterministic, we first define a generalized inverse mapping:

$$T^{-1}(s' \mid s, g) = a, \quad \text{where } T(a \mid s, g) = s'. \tag{28}$$

Then we could construct a policy $\pi_k^\tau(a \mid s, g)$:

$$\pi_k^\tau(a \mid s, g) = \begin{cases} \tilde{p}(s' \mid s, g), & \text{if } a = T^{-1}(s' \mid s, g), \\ 0, & \text{otherwise.} \end{cases} \tag{29}$$

$\square$

## B.2. Proof of Theorem 3.3

*Proof.*

$$\zeta_k(s, a) = |V_k(s, g) - V^*(s, g)| \tag{30}$$
$$= |V_k(s, g) - \mathcal{T}V_{k-1}(s, g) + \mathcal{T}V_{k-1}(s, g) - V^*(s, g)| \tag{31}$$
$$\leq |V_k(s, g) - \mathcal{T}V_{k-1}(s, g)| + |\mathcal{T}V_{k-1}(s, g) - V^*(s, g)| \tag{32}$$
$$= |V_k(s, g) - \mathcal{T}V_{k-1}(s, g)| + \gamma \left| \max_{a'} V_{k-1}(s', g) - \max_{s'} V^*(s', g) \right| \tag{33}$$
$$\leq |V_k(s, g) - \mathcal{T}V_{k-1}(s, g)| + \gamma \max_{s'} |V_{k-1}(s', g) - V^*(s', g)| \tag{34}$$
$$= \delta_k(s, g) + \gamma \max_{s'} \zeta_{k-1}(s', g), \tag{35}$$

where the inequality in (33) holds due to the triangle inequality and the inequality in (35) holds because the absolute difference between two maxima is bounded by the maximum of the absolute differences. $\square$

## B.3. Proof of Theorem 4.1

*Proof.* By setting the derivative of Equation (8) to 0, we obtain the following expression for $\hat{V}_{k+1}$ in terms of $\hat{V}_k$:

$$\forall s, g \in \mathcal{D}, k, \quad \hat{V}^{k+1}(s, g) = \hat{\mathcal{B}}^\pi \hat{V}^k(s, g) - \eta \frac{\mu(g|s)}{p_m^\alpha(g|s)}. \tag{36}$$

Since, $\mu(g|s) \geq 0, \eta \geq 0, p_m^\alpha(g|s) \geq 0$, we conclude that at each iteration, the updated V-value remains underestimated, i.e. $\hat{V}^{k+1} \leq \hat{\mathcal{B}}^\pi \hat{V}^k$. As proved in prior work (Kumar et al., 2020), if the reward function $r(s, g)$ and the transition function $T(s'|s, g)$ satisfy concentration properties, then with high probability(w.h.p) $\geq 1 - \delta, \delta \in (0, 1)$, the empirical Backup operator is bounded:

$$\forall V, s, g \in \mathcal{D}, \quad |\hat{\mathcal{B}}^\pi \hat{V}^k(s, g) - \mathcal{B}^\pi \hat{V}^k(s, g)| \leq \frac{C_{r,T,\delta} R_{\max}}{(1-\gamma)\sqrt{|\mathcal{D}|}}. \tag{37}$$

So the fixed point of Equation (36) is given by:

$$\hat{V}^\pi(s, g) \leq (I - \gamma P^\pi)^{-1} \left[ R - \eta \frac{\mu}{p_m^\alpha} + \frac{C_{r,T,\delta} R_{\max}}{(1-\gamma)\sqrt{|\mathcal{D}|}} \right]$$
$$= V^\pi(s, g) - \eta \left[ (I - \gamma P^\pi)^{-1} \left[ \frac{\mu}{p_m^\alpha} \right] \right] (s, g) + \left[ (I - \gamma P^\pi)^{-1} \frac{C_{r,T,\delta} R_{\max}}{(1-\gamma)\sqrt{|\mathcal{D}|}} \right] (s, g), \tag{38}$$

thus proving the underestimation of the value iterated via Equation (8). $\square$

Furthermore, when $\mu(g|s) = p_{\text{uni}}^{\mathcal{D}}(g)$ meaning the goals are sample uniformally from dataset $\mathcal{D}$ and the emprical Bellman operator is unbiased, the estimated value function $\hat{V}^{\pi}$ is as follows :

$$\hat{V}^{\pi}(s, g) = V^{\pi}(s, g) - \eta \left[ (I - \gamma P^{\pi})^{-1} \left[ \frac{p_{\text{uni}}^{\mathcal{D}}}{\alpha p_{\text{traj}}^{\mathcal{D}} + (1 - \alpha) p_{\text{uni}}^{\mathcal{D}}} \right] \right] (s, g). \tag{39}$$

## B.4. Proof of Proposition 4.2

*Proof.* For the state-goal pair $(s^+, g)$ which within the same trajectory, we have:

$$\hat{V}^{\pi}(s^+, g) = V^{\pi}(s, g) - \eta \left[ (I - \gamma P^{\pi})^{-1} \left[ \frac{1}{\alpha(\frac{p_{\text{traj}}^{\mathcal{D}}}{p_{\text{uni}}^{\mathcal{D}}} - 1) + 1} \right] \right] (s, g). \tag{40}$$

Obviously, $p_{\text{traj}}^{\mathcal{D}} \geq p_{\text{uni}}^{\mathcal{D}}$, then:

$$\hat{V}^{\pi}(s^+, g) \geq V^{\pi}(s^+, g) - \eta \left[ (I - \gamma P^{\pi})^{-1} \right] (s^+, g). \tag{41}$$

To guarantee

$$\hat{V}^{\pi}(s^+, g) \geq V^{\pi}(s^+, g) - \epsilon,$$

it suffices to ensure, for the given pair $(s^+, g)$,

$$\eta \cdot \left[ (I - \gamma P^{\pi})^{-1} [\mathbf{1}] \right] (s^+, g) \leq \epsilon.$$

Hence, one can choose

$$\eta \leq \frac{\epsilon}{\left[ (I - \gamma P^{\pi})^{-1} [\mathbf{1}] \right] (s^+, g)}.$$

If the inequality must hold for all state-goal pairs $(s, g)$, then

$$\eta \leq \frac{\epsilon}{\sup_{(s,g)} \left[ (I - \gamma P^{\pi})^{-1} [\mathbf{1}] \right] (s, g)} = \frac{\epsilon}{\left\| (I - \gamma P^{\pi})^{-1} [\mathbf{1}] \right\|_{\infty}}.$$

$\square$

## B.5. Proof of Proposition 4.3

*Proof.* For the state-goal pair $(s^+, g)$ which within the same trajectory, we have:

$$\hat{V}^{\pi}(s^+, g) \geq V^{\pi}(s^+, g) - \eta \left[ (I - \gamma P^{\pi})^{-1} \right] (s^+, g). \tag{42}$$

For the state-goal pair $(s^+, g)$ which is unconnected, $p_{\text{traj}}^{\mathcal{D}} = 0$, so we have:

$$\hat{V}^{\pi}(s^-, g) = V^{\pi}(s, g) - \eta \left[ (I - \gamma P^{\pi})^{-1} \left[ \frac{1}{1 - \alpha} \right] \right] (s, g). \tag{43}$$

We want to show that there exists a choice of $\alpha \in (0, 1)$ such that

$$\hat{V}^{\pi}(s^-, g) < \hat{V}^{\pi}(s^+, g) - \epsilon.$$

Recall the two key inequalities/equalities:

$$(1) \quad \hat{V}^{\pi}(s^+, g) \geq V^{\pi}(s^+, g) - \eta \left[ (I - \gamma P^{\pi})^{-1} \right] (s^+, g),$$

$$(2) \quad \hat{V}^{\pi}(s^-, g) = V^{\pi}(s, g) - \eta \left[ (I - \gamma P^{\pi})^{-1} \left[ \frac{1}{1 - \alpha} \right] \right] (s, g).$$

For brevity, define:

$$c_1 = \left[(I - \gamma P^\pi)^{-1}\right](s^+, g), \qquad c_2(\alpha) = \left[(I - \gamma P^\pi)^{-1}\left[\tfrac{1}{1-\alpha}\right]\right](s, g).$$

Then from (1) and (2) we get:

$$\hat{V}^\pi(s^+, g) \geq V^\pi(s^+, g) - \eta\, c_1, \qquad \hat{V}^\pi(s^-, g) = V^\pi(s, g) - \eta\, c_2(\alpha).$$

Subtracting the second from the first,

$$\hat{V}^\pi(s^+, g) - \hat{V}^\pi(s^-, g) \geq \left[V^\pi(s^+, g) - \eta\, c_1\right] - \left[V^\pi(s, g) - \eta\, c_2(\alpha)\right],$$

i.e.,

$$\hat{V}^\pi(s^+, g) - \hat{V}^\pi(s^-, g) \geq \left(V^\pi(s^+, g) - V^\pi(s, g)\right) + \eta\left[c_2(\alpha) - c_1\right].$$

As $\alpha \to 1^-$, the term $\frac{1}{1-\alpha}$ grows unboundedly. Under mild assumptions (nonnegative transitions, $\gamma < 1$, etc.), the operator $(I - \gamma P^\pi)^{-1}[\cdot]$ is monotone with respect to its input. Hence $c_2(\alpha)$ can be made arbitrarily large by choosing $\alpha$ close enough to 1. Therefore, $\eta\left[c_2(\alpha) - c_1\right]$ can exceed any fixed constant (including $\epsilon - \left(V^\pi(s^+, g) - V^\pi(s, g)\right)$, if necessary). Consequently, we can select $\alpha$ sufficiently close to 1 such that:

$$\hat{V}^\pi(s^+, g) - \hat{V}^\pi(s^-, g) > \epsilon.$$

Equivalently,

$$\hat{V}^\pi(s^-, g) < \hat{V}^\pi(s^+, g) - \epsilon.$$

Thus, for any fixed $\eta$, there always exists an $\alpha$ that satisfies the desired inequality.

$\square$

## B.6. Proof of Proposition 4.5

*Proof.* Any connected state-goal pair can be categorized into two cases:

**Case 1: In-trajectory state-goal pairs.**  According to Proposition 4.3, the proposition holds directly.

**Case 2: Cross-trajectory state-goal pairs.**  Assume $s^-$ and $g$ are in different trajectories within the dataset $\mathcal{D}$. By definition, $p^{\mathcal{D}}_{\text{traj}}(s^-, g) = 0$, and thus we have:

$$\hat{V}^\pi(s^-, g) = V^\pi(s, g) - \eta\left[(I - \gamma P^\pi)^{-1}\left[\tfrac{1}{1-\alpha}\right]\right](s, g).$$

On the other hand, suppose there exist $n$ intermediate states $s_1, s_2, \ldots, s_n$ connecting some $s^+$ to the same goal $g$, where each pair $(s_i, s_{i+1})$ is in-trajectory. By the properties of the quasimetric, we know:

$$\hat{V}^\pi(s^+, g) \geq \hat{V}^\pi(s^+, s_1) + \hat{V}^\pi(s_1, s_2) + \cdots + \hat{V}^\pi(s_n, g).$$

Therefore, to establish a strict value separation, it suffices to show:

$$\hat{V}^\pi(s^-, g) \leq \hat{V}^\pi(s^+, s_1) + \hat{V}^\pi(s_1, s_2) + \cdots + \hat{V}^\pi(s_n, g) - \epsilon. \tag{44}$$

According to the proof in Appendix B.5, it is straightforward to show that there exists some $\alpha$ such that the inequality in Equation (44) holds. Combining **Case 1** and **Case 2**, the proposition is proved.

$\square$

## C. Implementation Details

### C.1. Interval Quasimetric Embeddings

IQE trains an encoder $f(x; \theta)$ that maps states $s \in \mathcal{S}$ to latent vectors $z_s \in \mathcal{Z}$, alongside a latent quasimetric head $d_{\text{latent}}(z_s, z_g; \psi)$ to estimate the distance between state $s$ and goal $g$ in the latent space. Together, these components define a parameterized quasimetric over the state space $\mathcal{S}$:

$$d(s, g; \theta, \psi) := d_{\text{latent}}(f(s; \theta), f(g; \theta); \psi). \tag{45}$$

IQE structures input latents as two-dimensional matrices by reshaping. Given latents $u, v \in \mathbb{R}^{k \times l}$, IQE quantifies the Lebesgue measure of unions of multiple intervals along the real line:

$$\forall i = 1, 2, \ldots, k, \quad d_i(u, v) \triangleq \left| \bigcup_{j=1}^{l} [u_{ij}, \max(u_{ij}, v_{ij})] \right|. \tag{46}$$

A fundamental IQE is obtained by summing over all components:

$$d_{IQE\text{-}sum}(u, v) \triangleq \sum_{i=1}^{k} d_i(u, v). \tag{47}$$

A more flexible alternative, IQE-maxmean, introduces an additional parameter $\alpha$, blending maximum and mean reductions:

$$d_{IQE=maxmean}(u, v; \alpha) \triangleq \text{maxmean}(d_1(u, v), \ldots, d_k(u, v); \alpha) \tag{48}$$

$$\triangleq \alpha \cdot \max(d_1(u, v), \ldots, d_k(u, v)) + (1 - \alpha) \cdot \text{mean}(d_1(u, v), \ldots, d_k(u, v)). \tag{49}$$

### C.2. Implementation Details of CGCIVL

Our algorithm implementation is based on the reproduction of HIQL in the open-source OGbench and can be found at https://github.com/kkq2018/CGCIVL.git. On this basis, we implemented the quasimetric model based on IQE, denoted as $d_\theta$. The underestimation of the value function is represented as $\max d_{\theta_d}$. In order to stabilize the maximization process of $d_{\theta_d}$, a function $\phi$ is introduced, which is a monotonically increasing convex function. The final penalty term can be written as $\max \phi(d_\theta)$. At the same time, in the process of distilling the value function, a masking mechanism is employed to exclude certain loss contributions when the error between the predicted and target values falls within a specified range.

### C.3. Hyperparameter Settings

The detailed hyperparameter settings are shown in Table 2. The training process employed a batch size of 1024, with the policy and value networks designed as MLPs of dimensions (256, 256) and (512, 512, 512), respectively. The GELU activation function was used to ensure smooth gradient flow, while the Adam optimizer, configured with a learning rate of 0.0003, facilitated efficient parameter updates. To further stabilize the training process, the target network smoothing coefficient is set to 0.005.

*Table 2.* Hyperparameters.

| Hyperparameter | Value |
|---|---|
| Batch size | 1024 |
| Policy MLP dimensions | (256, 256) |
| Value MLP dimensions | (512, 512, 512) |
| Nonlinearity | GELU |
| Optimizer | Adam |
| Learning rate | 0.0003 |
| Target network smoothing coefficient | 0.005 |
| Penalty coefficient | 0.01 |

Some hyperparameters in the training process are tuned based on the specific task and environment. Specifically:

- In the antmaze-giant-stitch-v0 environment, the algorithm was trained for **2,000,000 steps**.

- In the humanoidmaze-giant-navigate-v0 and humanoidmaze-giant-stitch-v0 environments, the algorithm was trained for **3,000,000 steps**.

- For all other environments, the training steps were set to **1,000,000**, consistent with the settings in OGbench.

In the goal-stitching task, $\alpha$ was set to 0.5 to emphasize sampling from cross-trajectory pairs. In contrast, for the goal-navigating task, $\alpha$ was set to 0.9 to prioritize sampling from in-trajectory pairs. During the quasimetric distillation phase, another parameter, $\alpha_d$, was introduced to control sampling, ensuring the quasimetric focuses more on in-trajectory pair estimation. $\alpha_d$ is typically set to 0.8.

## D. Details of Environments and Datasets

We evaluate the proposed algorithm on OGbench, a recently introduced benchmark designed to assess offline goal-conditioned reinforcement learning algorithms across diverse tasks and datasets.

**Environments**    Our experiments involve six representative environments:

- *PointMaze:* A 2D navigation task where a point mass agent must traverse a maze to reach specified goal locations.

- *AntMaze:* A more challenging task requiring a quadrupedal ant agent with 8 degrees of freedom (DoF) to navigate complex environments.

- *HumanoidMaze:* The most complex task, involving a humanoid agent with 21 DoF, which requires high-dimensional control and advanced planning to solve.

- *Cube:* A pick-and-place manipulation task involving 1–4 cube blocks. The agent must learn to move, stack, swap, and permute cubes using a robot arm. Training data consists of play-style demonstrations with random pick-and-place actions. The task requires generalizable multi-object manipulation and long-horizon planning from unstructured data.

- *Scene:* A long-horizon manipulation task involving diverse objects such as a cube, drawer, window, and button locks. The agent must execute complex object interactions (e.g., unlocking and using containers) to reach goal configurations. It demands sequential reasoning and composition of atomic manipulation skills.

- *Puzzle:* A combinatorial reasoning task where a robot arm solves the "Lights Out" puzzle by pressing buttons on a grid (3×3 to 4×6). Pressing toggles adjacent buttons' states, and the goal is to match a target pattern. The agent must learn precise control and combinatorial generalization over a vast discrete state space, based on random-play trajectories.

Each environment is evaluated across three maze configurations:

- *Medium:* A small maze layout identical to the medium maze in D4RL  (Fu et al., 2020).

- *Large:* A larger maze with increased navigation complexity, matching the large maze in D4RL.

- *Giant:* A significantly larger maze, featuring paths up to 3000 steps, adapted from  Zhang et al. (2023) with added layout complexity.

**Datasets**    Three dataset types are used to evaluate different capabilities:

- *Navigate:* The dataset collected using a noisy expert policy that navigates the maze by sequentially reaching randomly sampled goals. This dataset is used to evaluate navigation performance.

- *Stitch:* The dataset composed of short goal-reaching trajectories. Completing tasks with this dataset requires stitching multiple short trajectories to reach the goal, testing long-horizon reasoning capabilities.

- *Play:* The dataset consists of "play"-style data collected by non-Markovian expert policies with temporally correlated noise. It features unstructured, exploratory interactions with the environment, where the agent performs random actions without specific task goals.

