# OpenReview forum: "Conservative Offline Goal-Conditioned Implicit V-Learning"
_ICML.cc/2025/Conference — ICML 2025 poster_

### Official Review · Reviewer_W7HN · 2025-02-17

**Overall Recommendation:** 3

**Summary:**

This paper proposes conservative goal-conditioned implicit V-learning (CGCIVL). The main insight of CGCIVL is to penalize cross-trajectory goal-conditioned values, which may potentially be overestimated, with a conservative regularizer. To improve the empirical performance of CGCIVL, the authors additionally employ other techniques (e.g., quasimetric value functions, hierarchical policy extraction, etc.) from the literature. They evaluate CGCIVL on OGBench, showing that it outperforms the previous methods on navigation environments, including those that require stitching.

**Claims And Evidence:**

The claims are empirically supported to some degree, but I do have several questions (see below).

**Essential References Not Discussed:**

I don't see any particular missing work.

**Experimental Designs Or Analyses:**

I don't have particular concerns about experimental designs or analyses other than the ones I listed in the weakness section below.

**Methods And Evaluation Criteria:**

Their evaluation criteria are reasonable in general, but the tasks are limited to (similar) navigation environments, and it'd have been more convincing if the authors had shown CGCIVL's performance on manipulation environments as well.

**Other Comments Or Suggestions:**

* $\mu$ is never formally defined (it is instead somewhat implicitly defined around L180). Relatedly, is $d^{\pi_\beta}$ correct? I suspect $s$ is sampled from the dataset distribution, not $d^{\pi_\beta}$ (note that they are different when dataset trajectories are truncated).
* I'd explicitly mention that $V_{\theta_d} \in \mathcal{Q}^-(\mathcal{S})$ around Equation (13). This is not explicitly stated in the current draft.
* What is the value of $\beta$ used for the experiments?

**Other Strengths And Weaknesses:**

### Strengths
* Figure 5 is quite convincing to me (especially in comparison with Figure 1). It is nice to see that $\alpha < 1$ improves performance on "stitch" datasets with the proposed techniques.
* CGCIVL achieves the best performance on almost all tasks employed in the paper.

### Weaknesses
* The paper omits a key ablation result -- how does CGCIVL's conservative regularization affect performance? This is the supposed key ingredient of the method, so I believe it is crucial to show how this design choice affects performance. In Figure 4 (in its current form), most of the performance gains are seemingly from quasimetric value functions and hierarchical policy extraction.
* The authors only evaluate CGCIVL on maze navigation environments. While the authors employ many datasets from OGBench, it'd have been much more informative if the authors had shown how CGCIVL works on other types of environments as well (e.g., manipulation). Does CGCIVL also work well on OGBench manipulation environments? If not, why?
* The authors use more training steps (e.g., 3M) for some challenging tasks (e.g., humanoidmaze), whereas the baseline results are obtained at 1M steps. Is CGCIVL still better than the baselines when they are trained with the same number of epochs?
* The proposed method is fairly complicated. It combines a number of different ingredients from previous methods -- quasimetric value functions, hierarchical policy extraction, implicit Q-learning, conservative Q-learning, etc. Hence, to some degree, their method is somewhat expected to work better than the baselines, because the baselines are usually more "atomic" (in the sense that they mostly employ one or two key techniques). While I don't think this is a major limitation, it would have been a great plus if their method had been simpler.

Overall, I'm not entirely convinced by the empirical results, mainly due to the lack of ablations and the limited types of environments. I'd be happy to adjust my score if these points are addressed.

**Questions For Authors:**

I don't have any questions other than the ones I asked above.

**Relation To Broader Scientific Literature:**

CGCIVL is built upon several existing methods --- IQL, GCIVL, HIQL, CQL, and QRL. Although the "novelty" of CGCIVL is not necessarily extremely prominent, I think the paper does have a reasonable degree of contribution (given that the claims are fully empirically supported).

**Theoretical Claims:**

I briefly reviewed the theoretical results (though I haven't thoroughly gone through the Appendix), and at least they look believable to me. The theoretical results are largely based on standard proof techniques about conservative value estimation.

---

> ### Author Rebuttal · Authors · 2025-03-30
>
> Thank you for your thoughtful review and valuable suggestions. We have carefully addressed each of your concerns in the responses below.
>
> ### R1: Methods and Evaluation Criteria
>
> We have extended our evaluation to manipulation environments (see Table 1 in the linked [PDF](https://anonymous.4open.science/r/additional-experiment-DDFC/icml_2025_rebuttal.pdf) for details). Detailed analysis is provided in our response R1 to Reviewer [96R5](https://openreview.net/forum?id=5ryn8tYWHL&noteId=wMyrrzRg7V).
>
> ### R2: Supplementary Material
>
> We would like to polish the README file to ensure our algorithm can be easily reproduced.
>
> ### R3: Weaknesses
>
> 1. We have conducted additional experiments comparing CGCIVL’s performance with and without the conservative regularization term (i.e., $\eta\neq 0$ and $\eta=0$ respectively). As shown in **Figure 1** (see the linked [PDF](https://anonymous.4open.science/r/additional-experiment-DDFC/icml_2025_rebuttal.pdf) for details), removing this component leads to a significant performance drop, confirming its critical role. Furthermore, we observe that the performance is robust within a suitable range of the regularization coefficient, but both excessively small and large values degrade the results.
> 2. See our response in R1.
> 3. In our paper, all algorithms were evaluated under the same number of training steps to ensure a fair comparison. Different from results in OGbench where baselines were trained with 1M steps, we trained all algorithms for longer steps in complicated environments (See Appendix C.3 for details). **Figure 3** (see the linked [PDF](https://anonymous.4open.science/r/additional-experiment-DDFC/icml_2025_rebuttal.pdf) for details) shows training curves for all algorithms in these complex environments.
> 4. We would like to clarify that CQL and quasimetric are the key techniques of our algorithm, addressing value overestimation on unconnected state-goal pairs. IQL is a necessary policy improvement technique, which could be replaced with other methods. The hierarchical structure is a common approach to handle long-horizon tasks and may be omitted in non-long-horizon environments.
>
> ### R4: Comments
>
> 1. $\mu(g|s)$ denotes an arbitrary distribution which satisfies $\operatorname{supp} \mu \subset \operatorname{supp} p_{m}^{\alpha}$. Trajectory truncation only alters the goal associated with states in different segments, without directly changing the distribution of states in the dataset. Therefore, we can approximate sampling states from $\mathcal{D}$ as sampling from $d^{\pi_{\beta}}$.
> 2. We'll mention $V_{\theta_d}\in\mathcal{Q}^-(S)$ near Eq. (13) for clarification in the revised manuscript.
> 3. The parameter $\beta$ in Equations (14)-(15) serves as the temperature coefficient for both the high-level and low-level policy extraction. We empirically set $\beta=3.0$ across all experiments.

---

> > ### Comment · Reviewer_W7HN · 2025-04-01
> >
> > Thank you for the detailed response. I appreciate the additional results, and they look convincing to me. I've raised my score to 3.
> >
> > Two minor comments:
> > * Why does Table 1 in the additional PDF not contain $\texttt{puzzle-3x3}$? In case this result is omitted because CGCIVL doesn't perform better: I believe a new method doesn't necessarily need to achieve the best performance on every single task. It'd be more informative to the community to present the entire result to enable a more holistic evaluation.
> > * $d^{\pi_\beta}(s)$ can be different from $\mathcal{D}$ even without goals, because the former is the discounted state marginal of the policy $\pi_\beta$ (with infinite rollouts), whereas the latter is the truncated state marginal distribution (e.g., consider the extreme case where every trajectory has length 1, in which case $\mathcal{D}$ would be the same as the initial state distribution, while $d^{\pi_\beta}(s)$ isn't).

---

> > > ### Author Response · Authors · 2025-04-02
> > >
> > > Thank you for your additional comments. As you suggested, we will include the full tasks in the puzzle environment in the final version of our paper. Regarding the sampling of $s$, I apologize for misunderstanding your point, and your review is correct. $d_{\pi_{\beta}}$ is indeed the state marginal distribution of the dataset and is related to $\pi_{\beta}$. Nevertheless, the conclusion of Theorem 4.1 still holds because $d_{\pi_{\beta}}$ is eliminated during the derivation and is not involved in the final expression.

---

### Official Review · Reviewer_uWae · 2025-03-10

**Overall Recommendation:** 3

**Summary:**

This paper introduces conservatism to prevent overestimation in unconnected state-goal pairs and uses a quasimetric value network to prevent underestimation in connected cross-trajectory state-goal pairs. Theoretical analysis is provided for the idealized version of the algorithm, and the practical implementation of the algorithm outperforms offline goal-conditioned RL baselines on OGBench.

**Claims And Evidence:**

The performance of the proposed CGCIVL algorithm is demonstrated through abundant experiments, which are convincing evidence.

However, the connection between theoretical analysis and the practical CGCIVL algorithm is weak, as the theory is based on an idealized version of CGCIVL (Eq. (8)). The theorems serve therefore more as a motivation than a guarantee.

**Essential References Not Discussed:**

No missing reference found.

**Experimental Designs Or Analyses:**

Experiments are solid to support the claim.

**Methods And Evaluation Criteria:**

Conservatism or regularization is a standard technique in reinforcement learning. The Quasimetric framework serves specifically for the goal-conditioned RL problem. Therefore, the proposed method is overall appropriate for the discussed problem.

The benchmark OGBench is suitable for goal-conditioned reinforcement learning.

**Other Comments Or Suggestions:**

1. All notations should be defined before using. Many notations are not standard across literatures, so they will cause confusion to readers without clear definition.
2. Algorithm 1 should be described in details to convince the readers that it can be implemented in practice. For instance, we need to discuss how to sample $g$.
3. Both Proposition 4.3 and 4.5 use $s^-$ and $s^+$, but they stand for different meanings in the two propositions. Therefore, we should consider using different notations.

**Other Strengths And Weaknesses:**

Strength: The paper investigates the advantage of combining conservatism and quasimetric.

Weakness:

1) Both conservatism and quasimetric are existing techniques in RL literature, although this does not severely harm originality, as the methods are tailored specifically for GCRL.
2) Lack of clarity. Many notations are used without formal definition (already discussed above). In addition, the main algorithm (Algorithm 1) needs detailed description. For instance, in Eq. (12), how do we estimate the expectation, and how to sample from $g$ from $p_m^\alpha(g|s)$?

**Questions For Authors:**

1. Does Theorem 4.1 hold for continuous state space or discrete state space? Similarly, does it require discrete action space or continuous action space also holds? This question reflects how the theoretical analysis aligns with practice.
2. It seems unnatural to require the value function to be a quasimetric, because sometimes the ground truth value function might not be a quasimetric. For instance, Suppose states A,B are connected, but both (A,C) and (C,B) are unconnected. Then we should have $V(A,B)=0$, $V(A,C)<0$, $V(C,B)<0$. This violates the quasimetric property.
3. Can CGCIVL fit in the framework of Eq. (8), assuming that we do not use any function approximation? This question determines the relationship between the theory and the practical algorithm.
4. Does Proposition 4.5 still hold if conservatism (regularization) is not added in the algorithm? This is relevant to the novelty of this paper.

I am willing to raise the score if the above concerns are resolved and the comments in the previous part are handled.

**Relation To Broader Scientific Literature:**

The two key components of CGCIVL, conservatism and quasimetric, are not novel in RL literature. The former is standard in RL algorithms e.g. CQL (Kumar et al., 2020), COMBO (Yu et al., 2021), and the latter is also proposed in https://arxiv.org/abs/2304.01203. The paper is only investigating the effect of combining these two techniques. Nonetheless, the successful combination of these two methods reveal the contribution of this paper.

**Theoretical Claims:**

Many notations are used without formal definition, thus hindering the understanding of the theorems along with their proofs. For instance, $\mu(g\mid s)$, $\hat V$ and $\hat{\mathcal{B}}$ in Eq. (8).

The formulation of Proposition 4.5 is problematic. As claimed in the proposition, the inequality should hold for any $\epsilon>0$, then this will imply that the $\hat V^\pi(s^-,g)$ has to be $-\infty$. However, this seems to be a minor typo.

I believe the theoretical claims are sound after the above issues are addressed.

---

> ### Author Rebuttal · Authors · 2025-03-30
>
> We sincerely appreciate your time and effort in reviewing our work. We have addressed each concern you raised below.
>
> ### R1: Claims and Evidence
>
> Thank you for the reviewer’s comment. The theoretical guarantees mentioned in our paper indeed refer to the algorithm prototype based on Eq. (8), rather than the practical algorithm. However, in Lemma 1 (Appendix A), we prove that the expectile regression in the practical algorithm is equivalent to the Bellman operator in Eq. (8). Additionally, compared to Eq. (8), the practical algorithm incorporates two key techniques: 1) hierarchical learning to address long-horizon tasks, and 2) quasimetric distillation to improve the efficiency of value learning. These techniques do not fundamentally alter the core components of the CQL-inspired penalty term and quasimetric, which form the foundation of the theoretical analysis. Therefore, while Eq. (8) does not exactly match the practical algorithm, we believe the practical algorithm still benefits from the theoretical guarantees established by the analysis.
>
> ### R2: Theoretical Claims
>
> 1. We clarify notations you mentioned as follows:
>    - $\mu(g|s)$ denotes an arbitrary distribution which satisfies $\operatorname{supp} \mu \subset \operatorname{supp} p_{m}^{\alpha}$.
>    - $\hat{V}$ represents an empirical estimate of the true value function $V$ during iteration.
>    - $\hat{B}$ denotes the empirical Bellman operator, which is the sample-based counterpart of the theoretical Bellman operator.
> 2. Yes. The correct statement of Proposition 4.5 should be: For any $\epsilon>0$ and $\eta>0$, there exists a hyperparameter $\alpha$ such that the inequality holds.
>
> ### R3: Relation to Broader Scientific Literature
>
> We would like to clarify that our work extends beyond a mere combination of existing techniques. The key contributions are:
>
> - Problem identification:
>
>   To the best of our knowledge, we are the first to formalize the critical issue of value overestimation for unconnected state-goal pairs in offline GCRL.
>
> - Feasible solutions:
>
>   Our solution penalizes the values of all cross-trajectory state-goal pairs while ensuring that values on connected pairs are not excessively under-estimated. We introduce a CQL-inspired regularization term for the first goal and use a quasimetric model for accurate value estimation of connected pairs to achieve the second. Both methods are supported by theoretical guarantees.
>
> - Difference with original methods
>
>   Unlike CQL, which penalizes OOD actions, we introduce a penalty term tailored for state-goal pairs. Unlike QRL, which trains value functions without value iteration, our approach incorporates quasimetric properties into the value iteration process to ensure accurate value estimation for connected state-goal pairs.
>
> The novelty of our method lies in re-engineering these components to address a new problem in offline GCRL, rather than merely combining them.
>
> ### R4: Weaknesses
>
> 1. The originality of our work is discussed in R3.
> 2. In R2.1, we provide explanations for any undefined notations and will include these details in the revised version of the paper. As defined in Sec 2, in Algorithm 1, states are randomly sampled, and goals are sampled in two ways: 1) $p_{rand}^{\mathcal{D}}(g)$ samples uniformly from all states in $\mathcal{D}$, and 2) $p_m^{\alpha}(g|s)$ samples from the same trajectory as state $s$ with probability $\alpha$, otherwise using $p_{rand}^{\mathcal{D}}(g)$.
>
> ### R5: Other Comments or Suggestions
>
> 1. See R2.1.
> 2. See R4.2.
> 3. Future revisions will implement distinct notation per proposition for clarity.
>
> ### R6: Questions
>
> 1. Although the proof in the original paper is based on discrete state and action spaces, its key components can also be  extended to continuous settings. Non-negative penalty terms for underestimation can be generalized to density-based terms, and concentration bounds for the empirical Bellman operator $\hat{B}^{\pi}$ do not require discretization. The Neumann series ensures that  $(I - \gamma P^{\pi})^{-1}$ remains well-defined in continuous spaces when $\gamma < 1$. Therefore, while Theorem 4.1 has not been strictly proven for continuous settings, our algorithm still benefits from the theoretical analysis in continuous environments, as further supported by experimental results.
> 2. As described in Sec 2 of our paper, the distance between state and goal should satisfies properties of quasimetric (Eq. (6)). However, the value function should exhibit an inverse relationship with the distance away from the goal (Eq. (7)). Thus we have $V(A;B)\geq V(A;C)+V(C;B)$, which is hold when $V(A;B)=0, V(A,C)<0,V(C,B)<0$.
> 3. Please refer to R1 where we discuss the differences between the practical algorithm and Eq. (8).
> 4. Proposition 4.5 is based on Theorem 4.1 by incorporating the quasimetric and replacing $\mu$ with the uniform distribution $p_{rand}^{\mathcal{D}}$. Consequently, Proposition 4.5 cannot hold if the conservatism is not included.

---

> > ### Comment · Reviewer_uWae · 2025-04-01
> >
> > Thank you for your response. It resolves my major concern of novelty, so I've updated my review and raised the score to 3.

---

### Official Review · Reviewer_96R5 · 2025-03-12

**Overall Recommendation:** 4

**Summary:**

This paper proposes a method for offline goal conditioned reinforcement learning with a penalty term to penalize the value function for unconnected state-goal pairs and does evaluation on OGBench. The results suggest the method outperforms previous methods on goal conditioned tasks.

**Claims And Evidence:**

The paper makes several claims.

1. Offline goal conditioned reinforcement learning suffers from value over estimation on unconnected state action pairs.

Support for this claim is presented through theoretical analysis in Theorem 3.3 and experimental evidence both in table 1 and figure 3.

2. The method proposed in this paper addresses the value over estimation and achieves better performance.

Generally this claim is supported from the main results in Table 1, but it only includes a subset of tasks from OGBench. The evidence would be more convincing if results are shown for all of the OGBench experiments.

**Essential References Not Discussed:**

Other related works that are essential are included in the paper.

**Experimental Designs Or Analyses:**

The experimental design is valid and the selection of benchmark is good. There are ablation studies to back up claims and compares against the other state of the art methods in offline goal conditioned reinforcement learning. The experimental section can be improved by providing results on the entire OGBench.

**Methods And Evaluation Criteria:**

The method and evaluation criteria appears to be well suited for the problem. The proposed method directly addresses the problem and the provides theoretical motivation. The benchmark selection is appropriate.

**Other Comments Or Suggestions:**

It would be helpful to include a mean for each of the environments.

**Other Strengths And Weaknesses:**

The paper is original and combines ideas in conservative value estimation to offline goal conditioned reinforcement learning. It is well motivated with its approach and achieves higher performance compared to previous algorithms.

A weakness of the paper is that it only compares in maze navigation tasks and is unclear how it would scale to other domains.

**Questions For Authors:**

1. Is there a reason the method is not run for all of the environments on OGBench?

2. The method has a lot of important hyperparameters. What are the sensitivity to other hyperparameters besides alpha?

**Relation To Broader Scientific Literature:**

The key contributions of the paper is related to the advancement of offline goal conditioned reinforcement learning. It proposes a new method that addresses an important problem in this direction of research.

**Theoretical Claims:**

The proofs appear to be correct.

---

> ### Author Rebuttal · Authors · 2025-03-30
>
> We sincerely appreciate your insightful feedback on our work. Please refer to detailed responses below to each of the raised concerns.
>
> ### R1: Weaknesses
>
> In order to provide more comprehensive evaluation of the performance, we have conducted additional experiments in three manipulation environments (Cube, Scene, Puzzle)  \- comprising a total of 8 manipulation tasks of varying complexity. The results, presented in **Table 1** (see the linked [PDF](https://anonymous.4open.science/r/additional-experiment-DDFC/icml_2025_rebuttal.pdf) for details), demonstrate that CGCIVL achieves superior performance in all manipulation tasks, particularly in scene and elementary cube tasks, consistent with results in maze environments. We will include these additional experimental results in the revised version of the paper.
>
> ### R2: Questions
>
> 1. In the original paper, we aimed to validate the algorithm in solving the goal stitching task. However, OGbench currently only provides the stitch dataset in maze environments. Nevertheless, we have supplemented our evaluation with additional experiments in manipulation environments to further verify the algorithm’s performance (See our response in **R1**). We will expand the experimental scope and incorporate these additional results into the final version of the paper.
> 2. Besides the analysis of $\alpha$, we have also performed sensitivity studies on both the penalty coefficient $\eta$ and the subgoal interval $k$. The analysis of $\eta$ and related experiments can be found in our response R2 to Reviewer [Q24N](https://openreview.net/forum?id=5ryn8tYWHL&noteId=56lsvAsJb7). **Figure 2** (see the linked [PDF](https://anonymous.4open.science/r/additional-experiment-DDFC/icml_2025_rebuttal.pdf) for details) shows the the performance of CGCIVL across different subgoal interval sizes. Results indicate that CGCIVL achieves the optimal performance with $k$ between $25$ and $50$. Overly small values of $k$ lead to the "signal-to-noise" issue in the value functions, as identified in the HIQL paper[1], while excessively large values of $k$ makes subgoals difficult to be achieved.
>
> ### R3:  Other Comments or Suggestions
>
>  We will include the mean score for each environment in the revised version of our paper to provide more comprehensive results.
>
> [1] Park, S. , Ghosh, D. , Eysenbach, B. , and Levine, S. Hiql: Offline goal-conditioned rl with latent states as actions. Advances in Neural Information Processing Systems,36, 2024b.

---

### Official Review · Reviewer_Q24N · 2025-03-13

**Overall Recommendation:** 3

**Summary:**

This paper proposes an algorithm for goal-conditioned offline RL called Conservative Goal-Conditioned Implicit V-Learning (CGCIVL). CGCIVL improves upon Hierarchical Implicit Q-Learning (Park et al., 2024b) by introducing two techniques. First, it adopts a regularizer similar to CQL (Kumar et al., 2020) to penalize values for unconnected state-goal pairs. Then, based on the observation that a goal-conditioned value function is a pseudometric, it models the value function with Interval Quasimetric Embeddings to prevent over-penalization of values for connected state-goal pairs. CGCIVL outperforms existing baselines on the OGbench(Park et al., 2024a) benchmark containing various goal-reaching tasks.

**Claims And Evidence:**

Most of the claims made in the submission are supported by clear and convincing evidence. For those that are problematic, refer to the following sections.

**Essential References Not Discussed:**

To the best of my knowledge, the paper has cited all of the essential references.

**Experimental Designs Or Analyses:**

The penalty coefficient $\eta$ also seems to play an essential role in the algorithm, but the authors have not conducted a sensitivity analysis on it.

**Methods And Evaluation Criteria:**

It is unclear why the authors use $V_{\theta_v}$ instead of the distilled $V_{\theta_d}$ to estimate the advantage functions $\tilde{A}_h$ and $\tilde{A}_l$. Aside from that, the proposed methods and the evaluation criteria make sense for the problem.

**Other Comments Or Suggestions:**

CQL adds a term to the loss function that maximizes the values for in-distribution data so that the regularizer is canceled out for in-distribution data. Similarly, adding a loss function term that maximizes the values for connected state-goal pairs might be helpful.

**Other Strengths And Weaknesses:**

Trajectory stitching is necessary for real-world problems because collecting high-quality data is challenging. This paper proposes an interesting method of applying HER for cross-trajectory state-goal pairs.

**Questions For Authors:**

I do not have any additional questions for the authors.

**Relation To Broader Scientific Literature:**

The proposed algorithm is mainly based on HIQL(Park et al., 2024b). The penalization term for unconnected state-goal pairs was inspired by CQL(Kumar et al., 2020). The observation that an optimal goal-conditioned value function is a quasimetric was proved by Liu et al. (2023). Finally, the authors modeled their value function using IQE(Wang & Isola, 2022a).

**Theoretical Claims:**

The $\alpha$ in Proposition 4.3 depends on the choice of the state-goal pairs, which means there might be no $\alpha$ that satisfies the condition for all state-goal pairs. The proposition becomes irrelevant since $\alpha$ is fixed for the entire training process. As Propositions 4.4 and 4.5 are all based on Proposition 4.3, the two propositions are also irrelevant.

---

> ### Author Rebuttal · Authors · 2025-03-30
>
> We sincerely appreciate the reviewer’s constructive feedback. Below, we respond to each concern point-by-point.
>
> ### R1: Methods and Evaluation Criteria
>
> The choice between using $V_{\theta_v}$ or the distilled $V_{\theta_d}$ for advantage estimation ($\tilde{A}^h$ and $\tilde{A}^h$) is flexible, as both approaches are empirically valid, surpassing all baselines. Our experiments confirm that $V_{\theta_d}$ achieves comparable performance when used for policy extraction, suggesting either value function can be adopted without compromising results. We will clarify this point in the final paper.
>
> |                         | pointmaze-large-navigate | pointmaze-giant-navigate | pointmaze-large-stitch | pointmaze-giant-stitch |
> | ----------------------- | ------------------------ | ------------------------ | ---------------------- | ---------------------- |
> | CGCIVL(with $\theta_v$) | $92 \pm 4$               | $80 \pm 12$              | $98 \pm 2$             | $81 \pm 17$            |
> | CGCIVL(with $\theta_d$) | $98 \pm 2 $              | $78 \pm 14$              | $96 \pm 6$             | $82 \pm 15$            |
>
> ### R2: Theoretical Claims
>
> Thank you for pointing out this question. Here we briefly explain why there exists  an $\alpha$ satisfies condition for all state-goal pairs.  Proof of Proposition 4.3 demonstrates that for any fixed $\epsilon > 0$ and arbitrary tuple $x=(s^+,s^-,g)$ sampled from dataset, where $(s^+,g)$ is in-trajectory and $(s^-,g)$ is cross-trajectory, there exists an $\alpha_x\in(0,1)$, such that inequality holds when $1>\alpha>\alpha_x$.  Consequently, let $\tilde{\alpha}=\sup_x{\alpha_x}$, then we could find a static $\alpha\in (\tilde{\alpha},1)$ , which satisfies the condition for all state-goal pairs. We will provide further clarification in the revised version of the paper.
>
> ### R3: Experimental Designs or Analyses
>
>  As suggested, we have conducted additional ablation studies to analyze the sensitivity of the penalty coefficient $\eta$, and results are presented in **Figure 1** (see the linked [PDF](https://anonymous.4open.science/r/additional-experiment-DDFC/icml_2025_rebuttal.pdf) for details).  The results indicate that both excessively small $\eta $ (causing insufficient regularization) and excessively large $\eta$ (over-constraining the optimization) degrade the performance. In practice, we determine the optimal $\eta$ through empirical validation across multiple candidate values.
>
> ### R4: Other Comments or Suggestions
>
> Maximizing values for connected state-goal pairs is indeed an interesting direction. However, we might need to carefully address several practical considerations: 1) directly sampling from the distribution of connected state-goal pairs in the dataset is difficult, and 2) further analysis is required to establish appropriate theoretical bounds, similar to those in CQL, when incorporating this approach. We plan to thoroughly explore these open issues in future work.

---

> > ### Comment · Reviewer_Q24N · 2025-04-06
> >
> > Thank you for your response. However, I still have a question to ask. The proof of Proposition 4.3 in the current version of the paper does not seem to mention the existence of a global upper bound $\alpha$ of $\alpha_x$. Could you elaborate on why such $\alpha$ should exist?

---

> > > ### Author Response · Authors · 2025-04-07
> > >
> > > Thank you for your additional comments. Proposition 4.3 indicates that for any $x = (s^+, s^-, g)$ , there exists an $\alpha_x$ such that the inequality holds. Furthermore, the conclusion of Proposition 4.3 also holds when $\alpha$ is greater than $\alpha_x$ based on the current proof process. Since the offline dataset is finite, there are only finite combinations of $x$. Therefore, we can select the maximum value from the finite set of $\alpha_x$ as the upper bound. Thus, in theory, we can use a fixed $\alpha$ that is not too small during training to achieve a lower estimate for the value of cross-trajectory state-goal pairs. We will include this clarification in the subsequent version of the paper.

---

### Decision · Program_Chairs · 2025-05-01

**Decision:**

Accept (poster)

**Comment:**

Authors present a way to incorporate stitching into hindsight experience replay in offline goal-conditioned RL with conservatism. This is a really interesting problem in goal-conditioned RL where there should be stitching in the goal space as well as the state space. While standard RL takes care of stitching in state space, hindsight experience replay does not allow for stitching in goal space. The authors propose to use quasimetric structure and conservatism to generalize to unconnected state-goal pairs.

There were concerns about completeness of the ablations, experiments, and incomplete definitions in the theorems. However, the authors provided additional results and clarifications in the rebuttal that assuaged most of the concerns. I recommend the authors make the suggested edits in the camera ready.